# A "scan and stop" assay identifies CD40 and CD70 as selective regulators of T-cell arrest on APCs

Vincent Gloe[1,2] , Richard Schregle[1], Christoph Ratswohl[1,3] , Jérémie Rossy[1,3]

**T-cell activation requires firm arrest on APCs, a process essential for effective clonal expansion and differentiation. Although the role of co-inhibitory signals and integrin-mediated adhesion in modulating T-cell arrest is established, the contribution of co-stimulatory molecules to this process remains poorly understood. Here, we developed a quantitative "scan and stop" assay using engineered CHO cells as minimalistic APCs to systematically assess the influence of co-stimulatory proteins on T-cell arrest. These APCs express only selected peptide–MHC complexes and co-stimulatory ligands, allowing controlled investigation of their roles in both naïve and experienced CD4+ and CD8+ T cells. We found that CD40 selectively promotes the arrest of pre-activated CD4+ T cells, whereas CD70 enhances the arrest of CD8+ T cells, correlating with expression patterns of their respective receptors, CD40L and CD27. High-resolution imaging further revealed mechanical deformation of APCs during synapse formation, suggesting force generation by T cells. Altogether, our results identify CD40 and CD70 as subtype-specific regulators of T-cell arrest and reveal a novel dimension in co-stimulatory control of immune synapse formation.**

## Introduction

T-cell activation represents a critical step in the initiation of the adaptive immune response. It leads to the clonal expansion of cytotoxic T cells, which eliminate virus-infected or tumour cells, and of helper and regulatory T cells, which coordinate the activity of a broad range of other immune cell types. Formation of the immunological synapse represents a crucial step in T-cell activation, characterised by the transition of T cells from active scanning to stable arrest upon encountering APCs such as dendritic cells (Dustin, 2009). How a T cell halts upon encountering an antigen can vary depending on signal strength and context (Friedman et al, 2005). Often, the transition from migration to arrest is remarkably rapid—within minutes of TCR engagement, a

T cell can go from continuous crawling to a complete stop (Dustin et al, 1997; Ozga et al, 2016). This abrupt halt suggests digital behaviour at the single-cell level when a sufficient TCR signalling threshold is reached.

Variations in how effectively T cells stop on APCs can profoundly influence the outcome of T-cell activation. Firm arrest, associated with a high-density or high-affinity antigen, generally drives a full activation program, typically characterised by calcium influx, transcriptional activation, clonal expansion, and differentiation into effector and memory cells (Dustin, 2009). Indeed, a few dendritic cells presenting a high amount of peptide–MHC can trigger immediate and prolonged T-cell arrest, resulting in productive activation (Henrickson et al, 2008). By contrast, transient stops or no stops at all, for example, upon encountering a low density of peptide, may lead to incomplete T-cell activation or defects in memory cell formation (Scholer et al, 2008). Hence, the probability of stopping can influence the efficiency of the immune response.

Stable T-cell arrest and synapse formation are accompanied by dramatic cytoskeletal remodelling. Upon TCR stimulation and integrin engagement, F-actin polymerisation surges at the T cell–APC interface, driving the T cell to spread over the APC and form a broad contact area (Dustin, 2009; Santos et al, 2016). In parallel with actin reorganisation, the microtubule cytoskeleton reorients to focus the T cell's secretory apparatus toward the immunological synapse (Combs et al, 2006). This cytoskeletal polarisation maintains functional polarity at the synaptic membrane (Compeer et al, 2018; Redpath et al, 2019) and is crucial for focused delivery of cytokines or lytic granules at the synapse by cytotoxic T cells (Griffiths et al, 2010).

Integrin-mediated adhesion, especially involving LFA-1 and its ligand ICAM-1, further stabilises T cell–APC conjugation. LFA-1 activation, regulated by "inside-out" signalling downstream of TCR engagement, promotes firm adhesion by preventing premature detachment (Dustin et al, 1997; Santos et al, 2016). The absence of ICAM-1 on dendritic cells notably impairs prolonged synapse formation, leading to compromised T-cell effector functions, such as reduced interferon-γ production (Scholer et al, 2008). In the absence of LFA-1 adhesion, T cells require orders-of-magnitude

[1]Institute of Cell Biology and Immunology Thurgau (BITG), University of Konstanz, Kreuzlingen, Switzerland   [2]Graduate School for Cellular and Biomedical Sciences, University of Bern, Bern, Switzerland   [3]Department of Biology, University of Konstanz, Konstanz, Germany

Correspondence: jeremie.rossy@gmail.com

 

more antigen to get activated, underscoring that tight adhesion lowers the threshold for TCR signalling (Katagiri et al, 2002). However, T-cell proliferation and differentiation into effector cells can occur without ICAM-1 (Feigelson et al, 2018).

Because it represents a critical step in T-cell activation, physical interactions between T cells and APCs can also be modulated by regulatory T cells (Tregs) to dampen or suppress conventional T-cell responses. In vivo, the absence of Tregs leads to antigen-specific T cells forming unusually long-lasting, stable contacts with dendritic cells, whereas the presence of Tregs keeps those contacts shorter and more transient (Tadokoro et al, 2006). Regulatory T cells also limit stable interactions between dendritic cells and low-avidity CD8[+] T cells by suppressing the production of chemokines, thereby reducing the probability and duration of T cell–DC contacts during priming (Pace et al, 2012). Mechanistically, Tregs' strong adhesion can induce changes in dendritic cells—for instance, causing cytoskeletal polarisation and a state of "contact-dependent lethargy" where the dendritic cell is less able to form productive synapses with other T cells (Chen et al, 2017). Like Tregs, regulatory macrophages can suppress T-cell priming by inhibiting stable dendritic cell–T-cell interactions through a nitric oxide-dependent mechanism (Yu et al, 2021).

Beyond the antigen receptor and integrins, various other receptors modulate T-cell stopping behaviour and synapse stability. Prostaglandin $E_2$ modulates synapse stability independently of proximal TCR signalling pathways, for example by affecting the Rap1 GTPase activity downstream of early signalling molecules such as ZAP70 and LAT (Wiemer et al, 2011). Co-inhibitory receptors further modulate T-cell arrest. CTLA-4 is up-regulated on activated T cells and competes with CD28 for B7 ligands. Engagement of CTLA-4 raises the threshold of TCR signalling required to maintain arrest, sending a "reverse stop signal" that increases T-cell motility and disrupts stable synapses (Schneider et al, 2006; Brunner-Weinzierl & Rudd, 2018). Signalling downstream of PD-1, another co-inhibitory receptor, also diminishes the strength of the stop signal, leading to less stable or shorter-lived contacts (Fife et al, 2009). In certain models, PD-1 engagement prevented full T-cell arrest, and blocking PD-1 allowed T cells to form longer interactions with antigen-bearing cells (Brunner-Weinzierl & Rudd, 2018).

In sum, fine-tuning the stop signal through stimulatory or inhibitory pathways, or via therapeutic interventions, can modulate immune responses by either enhancing T-cell activation, as in cancer immunotherapy where durable synapses with tumour cells are desirable, or by inducing tolerance, as in autoimmunity where reducing T-cell arrest on APCs may help dampen immune reactions. However, the role of co-stimulatory molecules in the regulation of T-cell arrest has not been deeply investigated.

In this study, we implemented a novel assay relying on CHO cells and transgenic OT-I and OT-II T cells to quantify the role of co-stimulatory molecules in the arrest of naïve and experienced CD4[+] and CD8[+] T cells. Because CHO cells do not express any co-stimulatory or co-inhibitory molecules that would activate mouse T cells, this approach allows us to evaluate the effect of the expression of a given co-stimulatory molecule on T-cell arrest and immunological synapse formation. We essentially focused our investigations on the TNF/TNFR superfamily members, which play a critical role in the communication between T cells and APCs

(Watts et al, 2025). Our results show that CD40 favours the arrest of CD4[+] T cells and that CD70 strongly increases the likelihood that CD8[+] T cells stop upon cognate peptide encounter. Higher magnification imaging further suggests that CD40L is present in endosomes at the rear and in the plasma membrane at the front of polarised migrating CD4[+] T cells, and that these cells exert a strong pushing force on APCs upon immunological synapse formation. Altogether, our data show a valid approach to quantify the role of surface molecules in T-cell arrest and uncover a novel function of CD40 and CD70 in CD4[+] and CD8[+] T-cell arrest, respectively.

# Results

## An in vitro "scan and stop" assay to investigate T-cell arrest on APCs

T-cell arrest on APCs relies on many factors, including the type of T cell, the extent of TCR triggering, and the microenvironment surrounding the APCs (biochemical cues, adhesive molecules, or other cell types), as well as co-stimulatory or inhibitory molecules expressed on the surface of the APCs. Intravital in vivo imaging provides the most physiological perspective on T-cell stopping. However, despite immense progress in genetic engineering, investigating the contribution of specific molecules to this process without altering other aspects of T-cell migration and activation remains technically challenging.

In vitro assays—typically using beads coated with stimulating anti-CD3 antibodies laid on an extracellular matrix component such as fibronectin—are more amenable to dissecting specific contributions to T-cell arrest. However, they do not involve antigen presentation by MHC molecules and require T cells to migrate on unphysiologically stiff substrates such as glass or plastic (Saitakis et al, 2017). More relevant paradigms, such as those involving BMDCs, also have limitations, as BMDCs express a wide range of chemokines, cytokines, and co-stimulatory/inhibitory molecules that intrinsically influence T-cell motility and activation. For these reasons, we sought to implement a "scan and stop" assay using: (a) APCs that (b) could be manipulated to express selected proteins of interest, and (c) a system where T cells would migrate over a homogeneous population of cells, only a subset of which would present a cognate antigenic peptide.

To meet these criteria, we used CHO cells engineered to express either single-chain OVA peptides in an MHC-I (OVA1) or MHC-II (OVA2) context to trigger activation and arrest of mouse OT-I CD8[+] T cells or OT-II CD4[+] T cells, respectively. The single-chain peptide-MHC (pMHC) molecules were co-expressed with the fluorescent protein iRFP713 using a P2A self-cleaving peptide sequence, enabling their identification during automated analysis. Notably, we verified that the expression levels of MHC-I and MHC-II on these CHO cells were comparable to those measured on mature BMDCs (Fig S1A and B). These minimalistic, activating APCs were plated on cover glasses to form a monolayer in a 1:15 ratio with non-activating CHO cells (CHO cells expressing the non-cognate VSV peptide in the pMHC-I context, or WT CHO cells for the pMHC-II

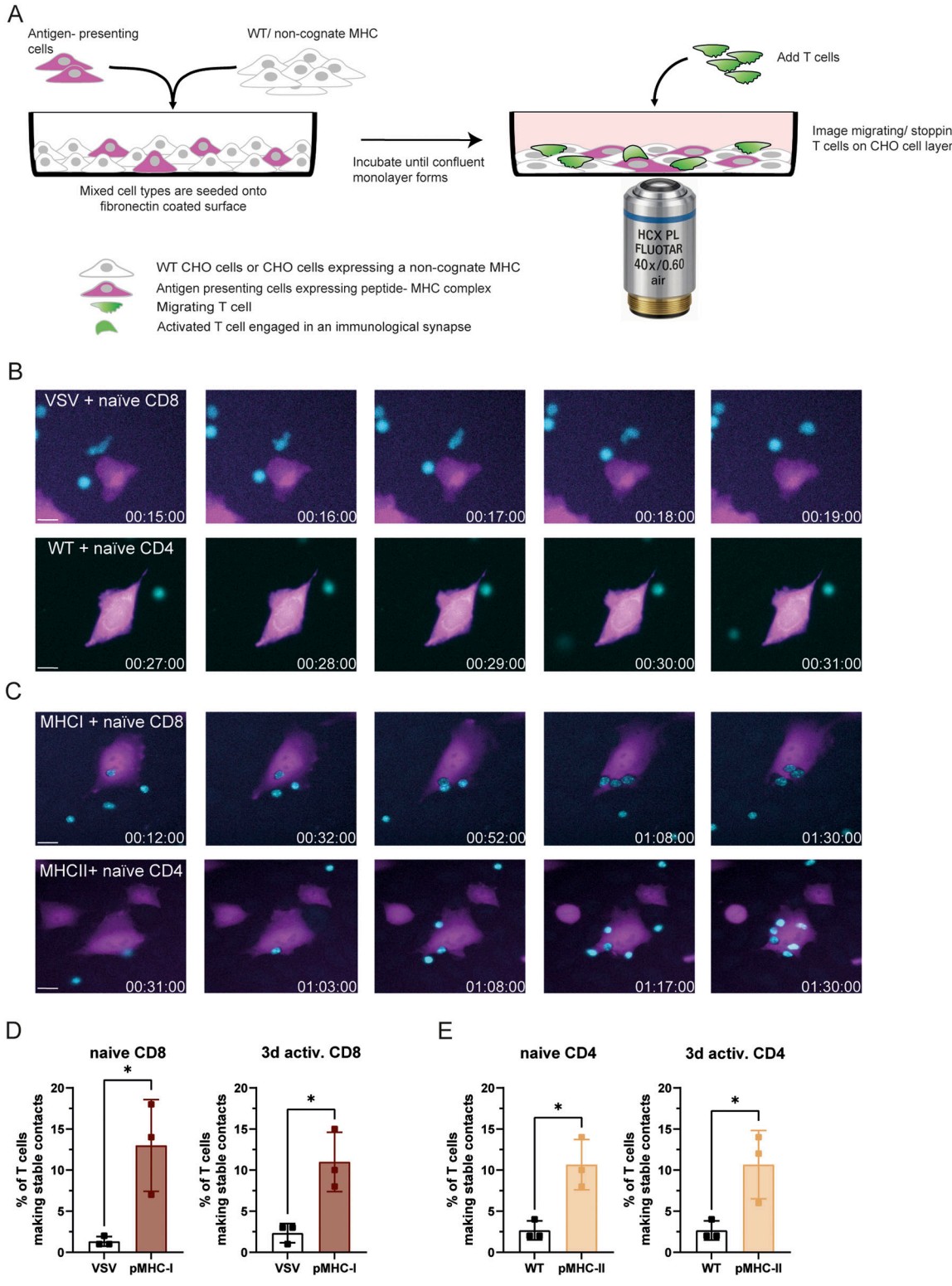

**Figure 1. An in vitro "scan and stop" assay to investigate T-cell arrest on APCs.**
**(A)** Schematic overview of the "scan and stop" assay. CHO cells expressing a single-chain OVA-peptide MHC-I (CHO-pMHC-I) or MHC-II (CHO-pMHC-II) complex and iRFP713 were mixed with either unstained CHO cells expressing an irrelevant single-chain MHC-I peptide complex (CHO-VSVg) or WT (CHO-WT) CHO cells. The mixed cell population was seeded into fibronectin-coated imaging chambers and cultured to form a confluent monolayer. OT-I CD8⁺ or OT-II CD4⁺ T cells, were then added. Epi-fluorescent time-lapse imaging was performed for 91 frames, with a frame time of 1 min. The outlines of pMHC-expressing CHO cells were identified using the ImageJ TrackMate plugin, and T cells were tracked using the same plugin. A custom analysis routine assessed T-cell tracks in relation to the underlying target cells. **(B)** Time-lapse images of naïve OT-I and OT-II T cells, stained with CellTracker Blue (shown in cyan), migrating on monolayers composed of either CHO-VSVg or CHO-WT cells. CHO-VSVg and CHO-WT cells were stained with CellTracker Deep Red (shown in magenta) and mixed with unstained CHO cells. Representative images illustrate that OT-I and

context; Fig 1A). Non-activating CHO cells did not express a fluorescent marker.

We used this system to investigate the arrest of CD4+ or CD8+ T cells that were either naïve or pre-activated with anti-CD3ε– and anti-CD28–coated beads for 3 or 5 d. T cells, labelled using Cell-Tracker Blue or Deep Red, were allowed to settle on the CHO monolayer for 15 min and imaged by epifluorescence microscopy (Leica DMi8) at 40x magnification and a frame rate of one image every 60 s (Fig 1A). Upon contact with the CHO monolayer, naïve T cells adopted a polarised morphology and motile behaviour. According to the assay's design, CD8+ T cells did not stop on non-activating CHO cells expressing the non-cognate pMHC-VSV (Fig 1B, Video 1). Similarly, CD4+ T cells failed to arrest upon contact with CHO cells expressing iRFP713 but no pMHC. In contrast, both T cell types rapidly arrested upon encountering activating CHO cells expressing the cognate single-chain pMHC (Fig 1C, Video 2).

To quantify the assay's output, we identified the surface areas and trajectories defined by the CellTracker-labelled T cells and the iRFP713-expressing CHO cells using the TrackMate ImageJ plugin (Schindelin et al, 2012; Tinevez et al, 2017, Fig S1C and D). Because the CellTracker signal did not fully cover the actual T-cell surface area once the images were made binary, we uniformly upscaled the polygons representing each T cell by 0.2%. We then applied two filtering steps. First, to remove dead T cells, we excluded every T-cell trajectory with a Euclidean distance (straight-line displacement) smaller than 15 $\mu$m—twice the diameter of an average T cell—based on the assumption that a living, motile T cell would move more than twice its diameter. Second, to remove non-adherent drifting T cells, we compared their trajectories with those obtained on non-adherent surfaces (uncoated or coated with 5% BSA; Fig S1E and F). Persistence above 0.5 was used as a cut-off to distinguish drifting from migrating cells and was applied to exclude non-adherent trajectories.

After filtering, the coordinates of polygons corresponding to T cells and CHO cells were processed using a custom HTML-based routine to calculate intersections between T cells and activating CHO cells for each frame. A "touch" was defined as any non-empty intersection between the polygon of a T cell and that of a target cell, with the area of intersection representing at least 10% of the T-cell surface. A "contact" was defined as a sequence of consecutive touches, including sequences with only a single touch. To account for brief instabilities such as flickering at the cell edge, a contact was retained even if up to five consecutive frames lacked an intersection. A "stable contact" was defined as a contact sequence that persisted until the end of the acquisition (90 min), as T cells interact for several hours with cognate APCs in lymph nodes (Stoll et al, 2002). Using these parameters, we calculated the percentage of stable contacts in the experiments shown in Fig 1B and C. To correct for minor variations in CHO cell density across experiments, the percentage of T cells forming stable contacts was

normalised using a density factor reflecting the local density of pMHC-I– or pMHC-II–expressing CHO cells relative to the group mean. Notably, >95% of the fluorescently labelled CHO cells expressed the single-chain OVA pMHC at their surface (Fig S1G).

The results confirmed the validity of our approach, as CD4+ and CD8+ T cells did not establish stable contacts with CHO cells expressing the fluorescent protein iRFP713 alone or a non-cognate pMHC (3.11% ± 1.27% for CD8+ T cells, 2.45% ± 0.99% for CD4+ T cells; Fig 1D and E). In contrast, 11.39% ± 3.95% of CD8+ T cells and 10.52% ± 3.23% of CD4+ T cells arrested on CHO cells expressing the cognate single-chain OVA pMHC. Surprisingly, pre-activation with antibody-coated beads did not affect the likelihood of CD4+ or CD8+ T cells forming stable contacts. These data further showed that CD4+ and CD8+ T cells had the same probability of establishing stable interactions.

## Pre-activation or subtype does not influence T-cell migration and arrest parameters

We then extended the number of parameters generated by our "scan and stop" assay to further explore the potential impact of pre-activation or T-cell type on motility and arrest. Any contact that did not persist until the end of the acquisition was defined as "transient." If a T cell was in contact with more than one target cell at the end of the acquisition, the longest contact was considered as stable, whereas the others were classified as transient. We also calculated the number of targets visited, defined as the total number of individual pMHC-expressing CHO cells that each T cell contacted, regardless of the number of interactions or whether the contact was stable or transient. Furthermore, for each stable contact, we measured the distance between the T cell and its position at the first touch for each frame during the interaction. We then recorded the maximum distance value and the time taken to reach it. The ratio of these two values was defined as the stopping speed, which describes how quickly a T cell comes to a complete halt during its contact with an APC.

To investigate whether a T cell's activation history influences its stopping behaviour, we generated distinct populations of antigen-experienced cells. To do this, T cells were pre-activated for 3 or 5 d with beads coated with antibodies against CD3ε and CD28. These populations were then compared with naïve T cells (day 0) in our "scan and stop" assay. Flow cytometric analysis confirmed that upon activation, both CD8+ and CD4+ T cells rapidly transitioned from a predominantly naïve phenotype to a memory-dominated composition, with effector/effector memory T cells peaking at around 20% and central memory T cells comprising the remaining 80% by day 3 and 5 (Fig S2A–C). T cells were put onto a monolayer of CHO cells, expressing iRFP713 and the corresponding antigenic pMHC (Fig 2A). Consistent with the data in Fig 1D and E, we observed

OT-II T cells do not interact with CHO-VSV or CHO-WT cells (Scale bar: 10 $\mu$m). **(C)** Time-lapse images of naïve OT-I and OT-II T cells, stained with CellTracker Blue (shown in cyan), migrating on monolayers containing CHO-VSVg cells mixed with CHO-pMHC-I, or CHO-WT cells mixed with CHO-pMHC-II. Representative images demonstrate that OT-I and OT-II T cells selectively interact with CHO cells expressing the corresponding peptide–MHC complex (Scalebar: 10 $\mu$m) **(D, E)** The percentage of naïve and anti-CD3/CD28 bead 3-d activated OT-I T cells engaging in stable contacts with CHO-VSV and CHO-pMHC-I (D) or naïve and 3-d activated OT-II T cells engaging in stable contacts with CHO-WT and CHO-pMHC-II cells **(A, E)** was quantified using the SaT assay, as described in (A). Shown are percentages (%) of T cells making stable contacts according to the defined criteria (see the Materials and Methods section). Statistical analysis was performed using unpaired, parametric t test. P < 0.05 (*).

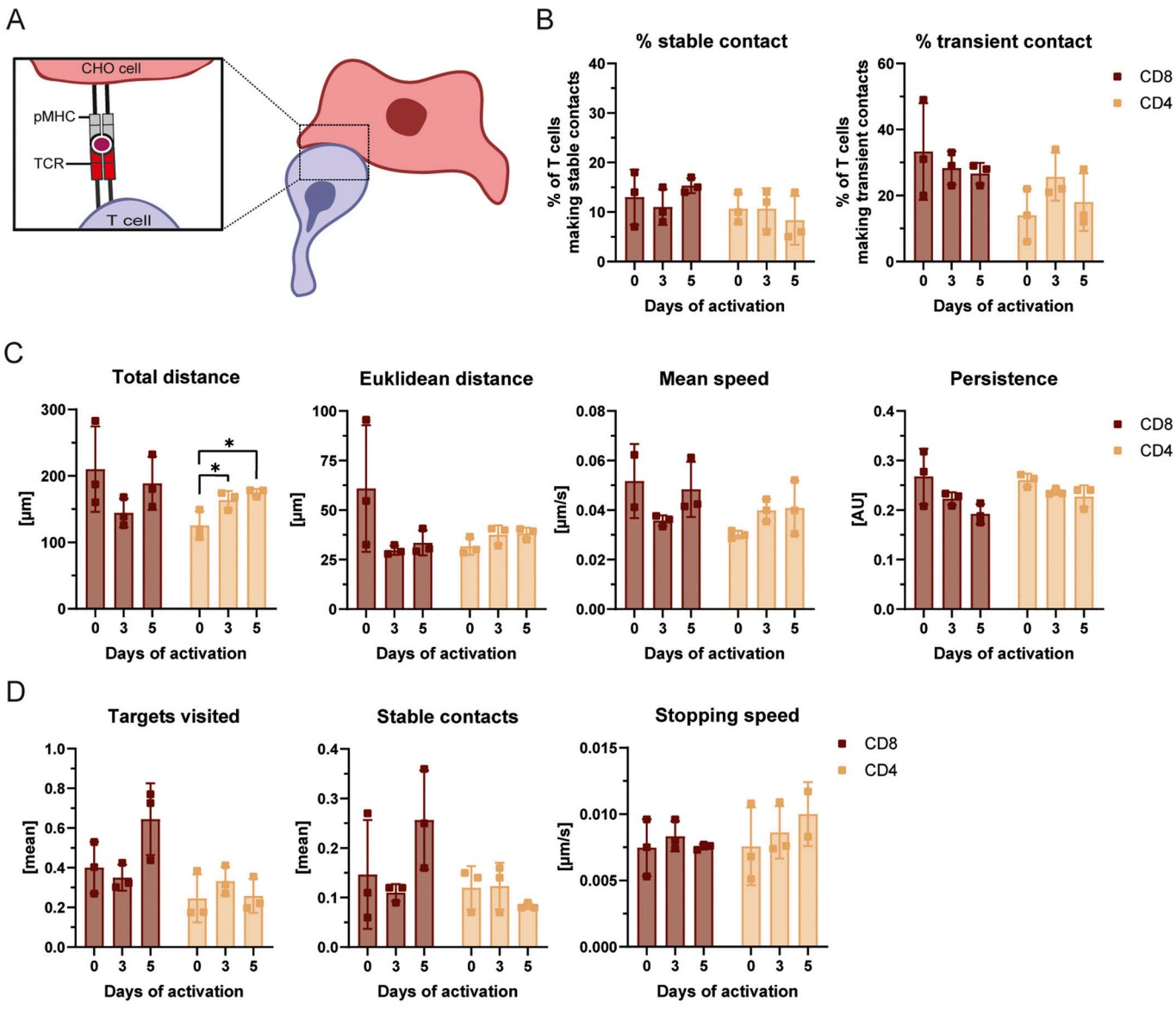

**Figure 2. Pre-activation or T-cell type does not influence migration and arrest parameters.**
**(A)** Schematic representation of the relevant interacting receptors of T (TCR) and CHO (pMHC) cells for T-cell activation and to study T-cell migration and arrest.
**(B)** Transient or stable contact formation, with CHO-pMHC-I or CHO-pMHC-II cells was assessed for naïve, 3-d-activated, and 5-d-activated OT-I and OT-II T cells using the scan and stop assay. Shown are percentages (%) of T cells making stable or transient contacts according to the defined criteria (see the Materials and Methods section). **(C)** Migration and arrest parameters as total and Euklidean distance ($\mu$m), mean speed ($\mu$m/s) and persistence (AU) of naïve, 3-d-activated, and 5-d-activated CD8$^+$ OT-I and CD4$^+$ OT-II T cells was measured. All parameters were calculated over the full track without excluding frames of contact via the TrackMate ImageJ plugin. Ordinary one-way ANOVA using Tukey's post hoc test was performed. $P < 0.05$ (*). **(D)** Mean number of target cells visited, mean number of stable contacts formed, as well as the stopping speed ($\mu$m/s) of naïve, 3-d activated, and 5-d-activated CD8$^+$ OT-I and CD4$^+$ OT-II T cells were determined via the TrackMate ImageJ plugin. For each stable contact, the distance between the T cell and its position at the initial touch was tracked across frames. The maximum displacement and the time taken to reach it were recorded, and their ratio was used to describe how rapidly a T cell came to a complete halt upon APC contact.

that CD4$^+$ and CD8$^+$ T cells had a similar capacity to arrest at APCs, and this was not altered by pre-activation (Fig 2B).

Interestingly, this equal capacity for arrest occurred despite differences in TCR expression. Flow cytometric analysis revealed that surface TCR levels were not down-regulated but were, in fact, modestly up-regulated on T cells pre-activated for 3 and 5 d compared with naïve T cells (Fig S2D and E). Furthermore, OT-I T cells displayed higher overall TCR levels than OT-II T cells, yet

both showed a similar probability of arrest. Taken together, these data suggest that once a sufficient number of TCRs are present, the absolute surface level is not the rate-limiting factor for initiating arrest in our system. Similarly, we detected no differences in their probability of making transient contacts, suggesting that there is no major difference in the motility of CD4$^+$ and CD8$^+$ T cells or naïve and memory T cells. Indeed, migration parameters generated by the TrackMate ImageJ plugin were broadly similar across all tested

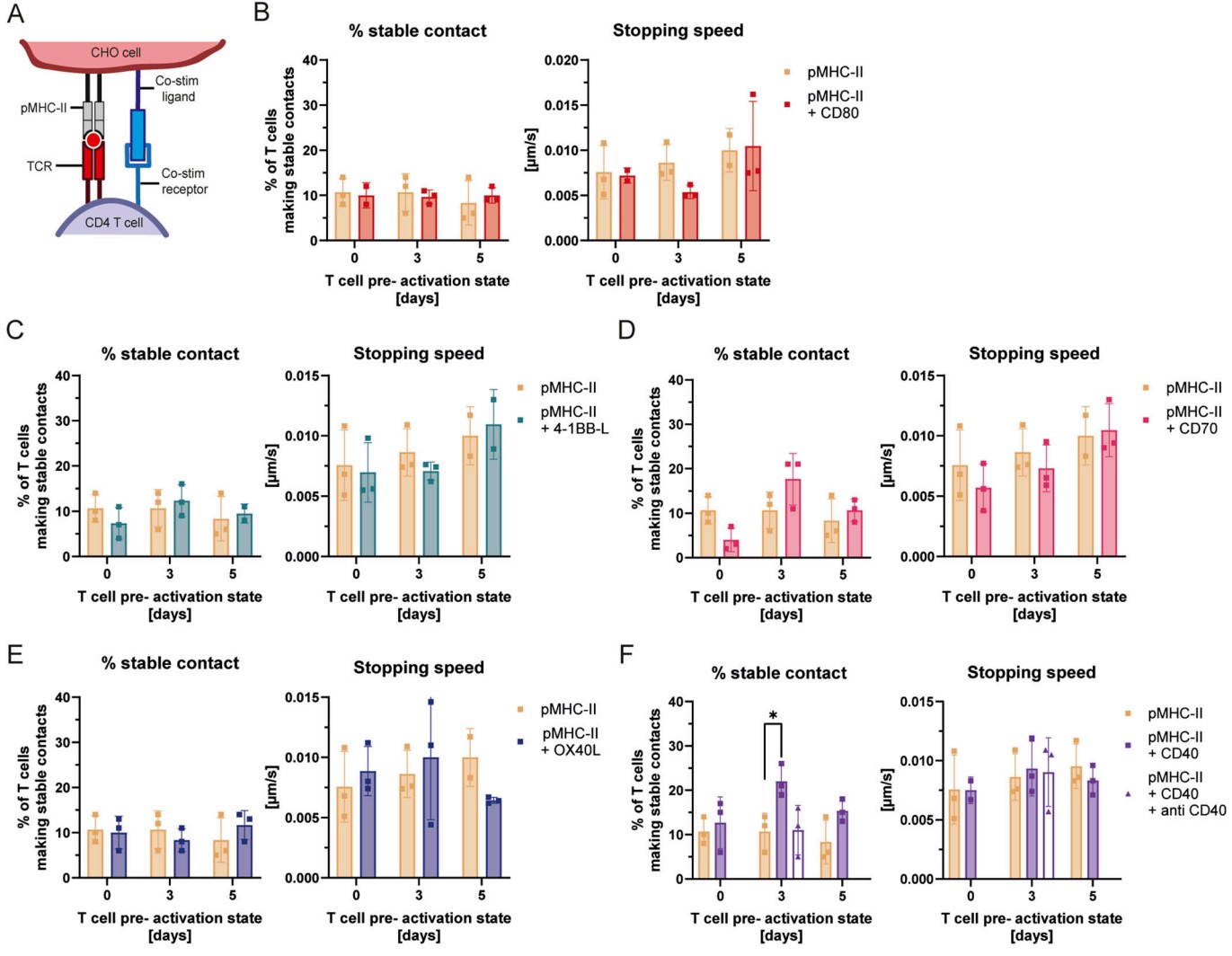

**Figure 3. Engagement of CD40 on CHO cells favours CD40L-mediated CD4⁺ T-cell arrest.**
**(A)** Schematic representation of the relevant interacting receptors of T cells and CHO cells for T-cell activation and further study of T-cell migration and arrest. **(B, C, D, E, F)** Stable contact formation and stopping speed of naïve, 3-d-activated, and 5-d-activated CD4⁺ OT-II T cells with CHO cells expressing pMHC-II alone or in combination with CD80 (B), 4-1BB-L (C), CD70 (D), OX40L (E), and CD40 (F) were assessed using the scan and stop assay. **(F)** also includes conditions where a CD40-blocking antibody was added to confirm specificity. Shown are the percentages (%) of T cells making stable contacts (left graphs) and their corresponding stopping speeds (right graphs) according to the defined criteria (see the Materials and Methods section). Statistical analysis was performed using unpaired, parametric $t$ tests comparing co-stimulatory–expressing pMHC-II-CHO cells with pMHC-II only at each given day, as well as comparing blockade conditions. $P < 0.05$ (*).

conditions, with the exception of the total distance travelled by CD4⁺ T cells (Fig 2C). Pre-activation increased the total distance travelled by CD4⁺ T cells, although this did not impact their capacity to locate or contact APCs. Accordingly, we observed no significant differences in the number of contacted pMHC-expressing CHO cells or in the total number of contacts with these cells. Finally, analysis of how T cells decelerated on pMHC-expressing CHO cells (stopping speed) also revealed no significant differences (Fig 2D). Altogether, our data show that, in the absence of adhesive or chemotactic cues, all T cells are equally capable of scanning for APCs. They further indicate that the capacity of the signal resulting from TCR triggering by cognate peptides presented on MHC molecules to induce T-cell arrest is similar across all tested T-cell types.

### Engagement of CD40 favours CD4⁺ T-cell arrest on APCs

Having observed that CD4⁺ T-cell stopping on APCs is not regulated by pre-activation alone, we sought to evaluate the influence of individual co-stimulatory pathways. To do this, we modified our minimalistic pMHC-II–expressing CHO cells to express a panel of classical co-stimulatory ligands, including CD80, a ligand for CD28, a co-receptor critical for T-cell activation (Esensten et al, 2016), as well as several members of the TNFR superfamily: 4-1BB-L, OX40L, CD70, and CD40 (Fig 3A).

Before assessing their functional impact on T-cell arrest, we validated that our engineered APCs expressed these ligands at physiologically relevant levels. As shown by flow cytometry in Fig S3, the surface expression of CD80, 4-1BBL, CD70, and CD40 on our

transduced CHO cells was comparable to the levels found on mature, LPS-stimulated BMDCs.

Having confirmed the physiological relevance of our system, we assessed the ability of these individual ligands to modulate CD4+ T-cell arrest. Despite its known role in actin cytoskeleton reorganisation during T-cell activation (Molon et al, 2022), the expression of CD80 on APCs did not affect the probability of CD4+ T-cell arrest (Fig 3B). Similarly, our screen of the other TNFR superfamily members, 4-1BB-L, OX40L, and CD70, showed no significant differences regarding the ability of motile CD4+ T cells to stop on APCs (Fig 3C–E).

By contrast, pMHC-expressing CHO cells co-expressing CD40 were more efficient at inducing arrest of CD4+ T cells pre-activated for 3 d (10.7 ± 3% versus 20.7 ± 4.5%, Fig 3F). To understand this effect, we measured the expression of CD40L, the ligand for CD40, on naïve and pre-activated T cells (Fig S4B). CD40L was most abundant on CD4+ T cells pre-activated for 3 d compared with day 0 and 5, thereby correlating with the enhanced arrest observed in this condition. To determine whether the enhanced arrest mediated by CD40 was a result of faster synapse formation, we analysed the stopping speed of T cells upon encountering CD40-expressing APCs. We found that CD40 expression did not alter the time it took for the antigen-specific T cells to arrest compared with pMHC-only conditions (Fig 3F). This indicates that this co-stimulatory interaction primarily modulates the probability and long-term stability of the arrest, rather than the initial deceleration kinetics, likely by lowering the signalling threshold required for T cells to commit to a stable synapse.

Finally, to confirm that the increased T-cell arrest was specifically driven by CD40 rather than non-specific adhesion to the engineered APCs, we introduced neutralizing antibodies into our co-culture system. The addition of a CD40-blocking antibody completely abrogated the enhanced stable contact formation of pre-activated CD4+ T cells induced by CD40-expressing APCs, reducing the arrest frequency to levels comparable to the pMHC-II only control (Fig 3F). This blockade experiment confirmed the specificity of our minimalistic APC system and established that direct receptor-ligand engagement of CD40 to CD40L is necessary to drive the enhanced T-cell arrest. Altogether, these data suggest that among classical co-stimulatory molecules involved in the activation of CD4+ T cells, CD40 specifically modifies the capacity of APCs to trigger the arrest of motile CD4+ T cells.

### Engagement of CD70 strongly promotes the arrest of motile CD8+ T cells

Following the same approach as for CD4+ T cells, we modified pMHC-I–expressing CHO cells to present co-stimulatory molecules of interest to CD8+ T cells (Fig 4A). As observed for CD4+ T cells, expression of CD80 and 4-1BBL had no effect on the probability of CD8+ T-cell arrest (Fig 4B and C). Unlike for CD4+ T cells, CD40 expression did not influence CD8+ T-cell arrest, suggesting that the effect observed in Fig 3 is specific to CD4+ T cells (Fig 4D). This difference could be explained by differential CD40L signalling in experienced CD4+ versus CD8+ T cells, or by a threshold effect linked to CD40L expression, as 3-d pre-activated CD4+ T cells

expressed 2.9 times more CD40L than CD8+ T cells after 3 d of pre-activation (Fig S4A–C). Nonetheless, the functional consequences of CD40L ligation by CD40 on T cells are not well understood, and further investigations would be required to clarify the difference in sensitivity between CD4+ and CD8+ T cells to CD40 expression on APCs.

Another classical co-stimulatory ligand of the TNFR superfamily, OX40L, had no impact on CD8+ T-cell arrest in our assay (Fig 4E), whereas CD70 did not influence CD4+ T-cell arrest; it was specifically included in the list of tested co-stimulatory molecules because of its role in CD8+ T-cell immunity. By engaging its receptor CD27, CD70 provides potent signals that promote T-cell survival, proliferation, and the development of long-term memory populations (Hendriks et al, 2000; Taraban et al, 2004). Given its essential role in determining the quality and durability of the CD8+ T-cell response, we hypothesized that the CD70⁻CD27 axis might also regulate the earliest stages of CD8+ T-cell priming by modulating the initial physical arrest. Indeed, pMHC-I–expressing CHO cells presenting CD70 at their surface were more than twice as efficient as pMHC-I-only–expressing CHO cells in promoting arrest of pre-activated CD8+ T cells (11 ± 2.9% versus 21.3 ± 3.8%, 15.3 ± 1.2% versus 35.7 ± 1.9%, Fig 4F). These findings are consistent with the expression pattern of the CD70 receptor, CD27. CD27 is expressed on naïve CD8+T cells and, as our data confirm, is strongly upregulated after 3 and 5 d of pre-activation with antibody-coated beads (Fig S4A and D). Similar to our observations with CD40, the presence of CD70 increased the overall frequency of stable contacts without accelerating the stopping speed of the T cells (Fig 4F).

Again, consistent with our findings for CD40, we verified the specificity of the CD70-mediated arrest using a similar blockade approach. Disruption of the CD70⁻CD27 axis with a CD70-neutralizing antibody completely abolished the enhanced stable contact formation of CD8+ T cells on CD70-expressing APCs (Fig 4F). This confirms that, much like the CD40⁻CD40L interaction, direct receptor-ligand engagement is strictly required to drive the enhanced arrest of CD8+ T cells. This suggests that the CD70⁻CD27 axis operates through a comparable mechanism, reducing the threshold for stable synapse commitment rather than altering the initial kinetics of the arrest.

CD70 is generally expressed at the surface of APCs, particularly dendritic cells, after their initial interaction with CD40L-expressing T cells, in a process known as ""licensing" (Laidlaw et al, 2016). Licensing enhances the capacity of dendritic cells to activate CD8+ T cells, notably through CD70 expression. Hence, our data suggest that one mechanism by which CD70-expressing dendritic cells support CD8+ T-cell priming may be through their enhanced capacity to arrest motile CD8+ T cells. This finding reinforces the idea that T-cell arrest plays an essential role in shaping the outcome of T-cell activation.

### High-magnification imaging identifies CD40L at the front of motile CD4+ T cells and reveals strong pushing forces upon immunological synapse formation

Microscopy acquisitions to evaluate the influence of co-stimulatory proteins on the percentage of T cells arresting at cognate APCs were performed at low magnification (40x) using an epifluorescence microscope. To confirm the presence of

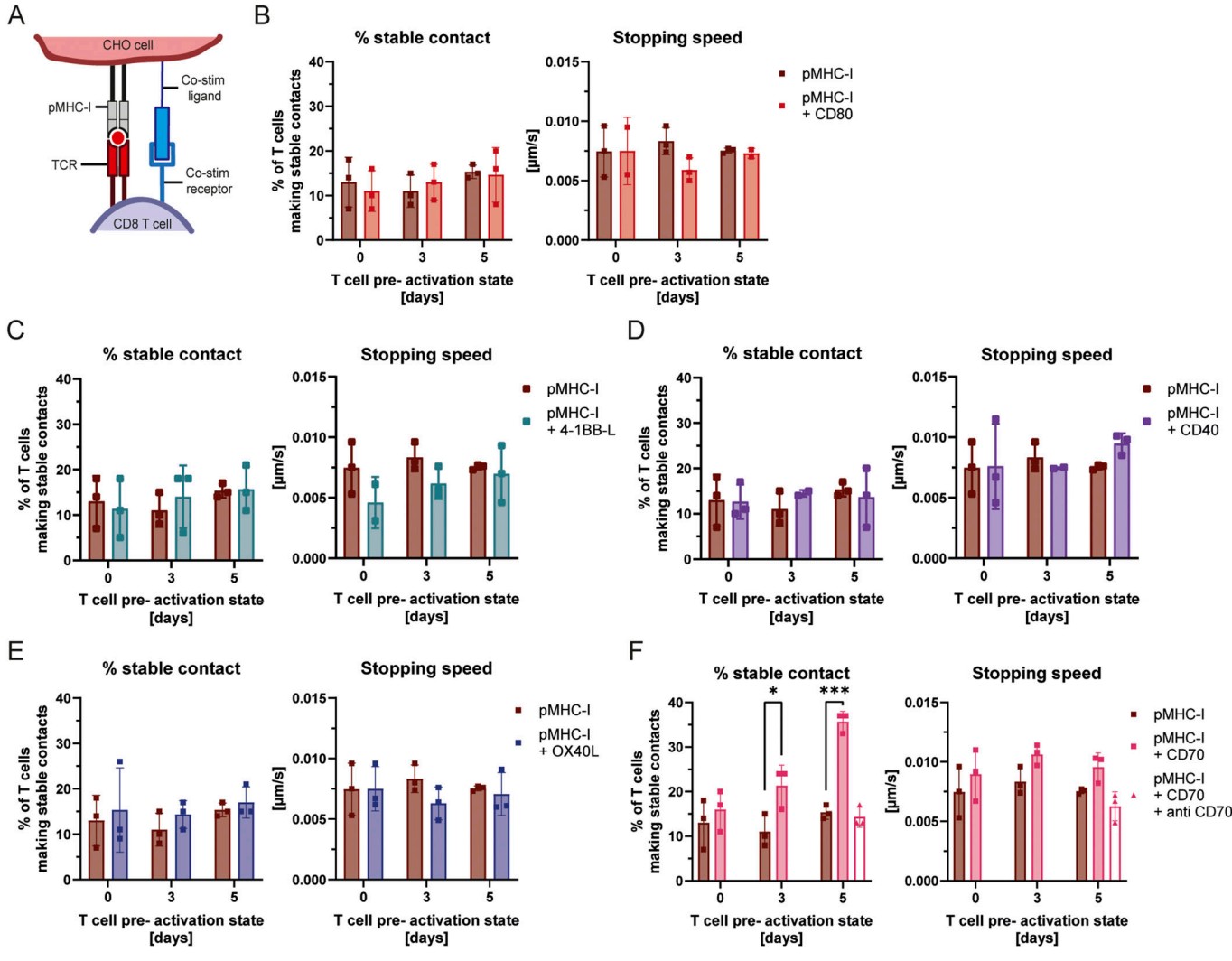

**Figure 4.   Engagement of CD70 on CHO cells via CD27 strongly promotes the arrest of motile CD8 T cells.**
**(A)** Schematic representation of the relevant interacting receptors of T cells and CHO cells for T-cell activation and further study of T-cell migration and arrest. **(B, C, D, E, F)** Stable contact formation and stopping speed of naïve, 3-d, and 5-d-activated CD8[+] OT-I T cells with CHO cells expressing pMHC-I alone or in combination with CD80 (B), 4-1BB-L (C), CD40 (D), OX40L (E), and CD70 (F) were assessed using the scan and stop assay. **(F)** also includes conditions where a CD70-blocking antibody was added to confirm specificity. Shown are the percentages (%) of T cells making stable contacts (left graphs) and their corresponding stopping speeds (right graphs) according to the defined criteria (see the Materials and Methods section). Statistical analysis was performed using unpaired, parametric *t* tests comparing co-stimulatory molecule-expressing pMHC-I-CHO cells with pMHC-I only at each given day, as well as comparing blockade conditions. $P < 0.05$ (*), $P < 0.001$ (***).

CD40L at the plasma membrane of motile CD4[+] cells, we imaged the same experimental setup using a high-resolution oil-immersion 60× 1.42 NA objective on a confocal microscope (Nikon Eclipse Ti2). To visualize CD40L at the synapse, we transduced pre-activated OT-II CD4[+] T cells with a retrovirus encoding CD40L-mCherry. We then used high-resolution confocal microscopy to image the interaction of these T cells with CHO-pMHC-II cells engineered to express the CD40 receptor. Full z-stacks of T cells contacting CD40-expressing pMHC-II CHO cells were recorded every 20 s with a resonant scanner. The images were then de-noised with an AI tool and deconvoluted using NIS-Elements (Nikon).

In polarised, migrating CD4[+] T cells, most of the CD40L signal was localised within endosomes at the rear end of the cell as

previously described (Koguchi et al, 2007) (Fig 5A, Video 3). However, CD40L was also detectable at the plasma membrane at the leading edge of polarised T cells, making it readily available to engage CD40 on CHO cells. As expected, once T cells established contact with pMHC-II-CHO cells, CD40L endosomes were translocated to the immunological synapse (Fig 5A, 10 min). On the APC side, CD40 was evenly distributed throughout the plasma membrane of CHO cells, and this distribution was not altered after T-cell contact, indicating that CD40L engagement alone is not sufficient to passively increase CD40 density at the synaptic membrane. However, we observed that T cells markedly deformed the pMHC-II-CHO cells, pushing deeply into the CHO cell body in more than half of the observed conjugates (Fig 5A and B). This is consistent with previous work showing that T cells exert

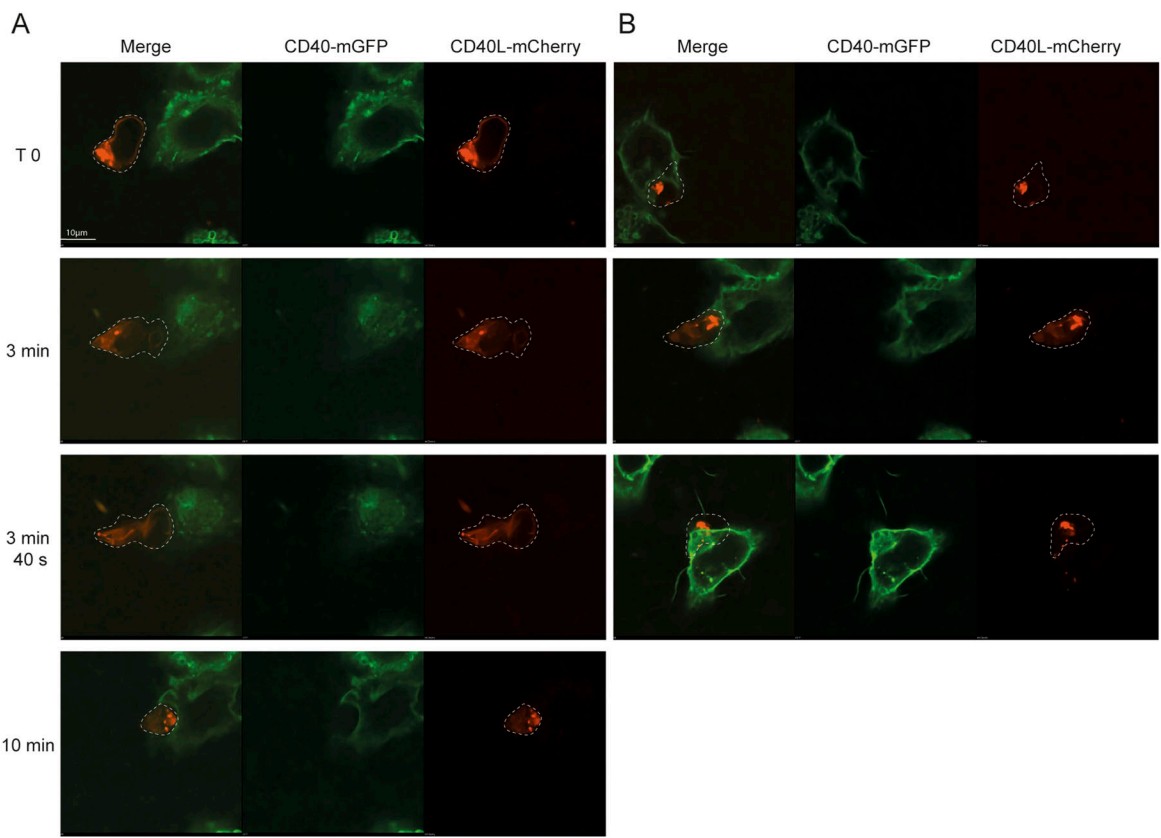

**Figure 5. High-magnification imaging of CD40L-mCherry–expressing T cells encountering CD40-mGFP–expressing pMHC-II-CHO cells for subsequent synapse formation.**
**(A, B)** CD4[+] OT-II T cells, transduced with a retroviral backbone encoding for CD40L–mCherry, were imaged together with CD40-mGFP pMHC-II–expressing CHO cells using the SaT assay. **(A)** Representative time-series images illustrate the initial contact between a CD4[+] T cell and a CHO cell (A). **(B)** Representative images illustrate the deformation of CHO cells by CD4 T cells (B). Images were acquired using confocal imaging with 60x magnification. **(A)** Z-stack compression in (A) was performed using NIS-Elements (Nikon).

strong mechanical force on APC (Hu & Butte, 2016; Rossy et al, 2018).

While electron microscopy has revealed that T cells extend finger-like projections to contact APCs (Leithner et al, 2021), such large-scale deformation has not been reported in professional APCs like dendritic cells. This suggests that physiological APCs may respond to T-cell engagement of pMHC by stiffening their cell bodies. Indeed, dendritic cell maturation is associated with increased cell stiffness, which enhances their ability to prime T cells (Blumenthal et al, 2020; Lühr et al, 2020; Leblanc-Hotte et al, 2023). Altogether, our high-magnification data show that CD40L is present at the front of polarised, motile CD4[+] T cells as they form immunological synapses with cognate APCs. Our findings further suggest that T cells apply strong mechanical forces toward APCs during synapse formation.

## Discussion

The arrest of motile T cells on APCs represents a critical step in T-cell activation, conditioning both the efficiency and outcome of clonal expansion and differentiation. Beyond the affinity and density of the presented cognate peptide, several factors regulate this transition, including regulatory T cells and macrophages, Prostaglandin $E_2$, and the co-inhibitory molecules CTLA-4 and PD-1. However, the role of co-stimulatory proteins in this process had not previously been investigated. Here, we developed a novel "scan and stop" assay to systematically evaluate the role of these proteins in the arrest of motile CD4[+] and CD8[+] T cells, both naïve and experienced. The assay relies on CHO cells used as minimalistic APCs, which intrinsically express no proteins capable of triggering or influencing T-cell arrest or immunological synapse formation, and which were engineered to express single-chain peptide–MHC complexes.

Interestingly, we observed that both naïve and highly preactivated T cells showed a similar propensity to arrest in our system. This robust arrest occurred despite our flow cytometry data showing that 3- and 5-d pre-activated T cells actually express higher absolute levels of surface TCR compared with naïve cells. However, considering that the TCR can be triggered by as few as a single pMHC complex (Irvine et al, 2002; Huang et al, 2013), the absolute surface level of TCR is likely not the sole decisive factor. Instead, our engineered APCs, which present a uniform, high

density of single-chain pMHC complexes, intentionally provide a saturating "Signal 1" that overcomes these variations in TCR expression and avidity. This standardizes the baseline of antigen recognition, allowing us to isolate and interrogate the specific contributions of "Signal 2" co-stimulatory molecules to the physical arrest process. However, while advantageous for this reductionist approach, the supraphysiological antigen presentation also constitutes a limitation of our assay. Because high levels of "Signal 1" can alter the native dynamics and downstream molecular kinetics of T-cell arrest, extrapolating these findings requires careful consideration and future validation under more physiological in vitro and in vivo settings. Our results show that neither the type of T cell nor its pre-activation status influences its capacity to migrate on a CHO cell monolayer, to locate APCs, or to arrest upon encountering a cognate antigenic peptide. The equal stopping capacity of CD4$^+$ and CD8$^+$ T cells can be explained by the fact that TCR affinities for both MHC classes generally fall within a similar range (6–10 $\mu$M), with no systematic difference between CD4$^+$ and CD8$^+$ T-cell receptors (Merwe & Davis, 2003). Similarly, the intrinsic affinity of the TCR is comparable between naïve and memory cells. Memory cells display higher avidity for cognate APCs, but this is largely due to enhanced TCR clustering (Fahmy et al, 2001) and increased expression of adhesion molecules (Berard & Tough, 2002), which should not contribute to T-cell arrest in our system. Indeed, TCR clustering occurs after immunological synapse formation (Grakoui et al, 1999). Our minimalistic system allows for the specific interrogation of TCR and co-stimulatory signals in the absence of confounding adhesion pathways. Our engineered CHO cells do not express any murine adhesion molecules, and while they express endogenous hamster ICAM-1, the low sequence identity (72%) to its mouse counterpart makes it unlikely to support a significant adhesive interaction with murine LFA-1. Therefore, the enhanced arrest we observe is unlikely to be driven through a canonical integrin-adhesion axis. This further supports a model where co-stimulation directly lowers the TCR signalling threshold to initiate the arrest program. Hence, our data indicate that all T cells are equal in their capacity to arrest at APCs and suggest that any differences observed between CD4$^+$ and CD8$^+$ T cells, or between naïve and antigen-experienced cells, are primarily due to the broader cellular context of these interactions.

Expression of co-stimulatory molecules on minimalistic CHO-based APCs further revealed that most tested co-stimulatory molecules, including CD80, had no impact on T-cell arrest—except for CD40, which promoted the arrest of CD4$^+$ T cells pre-activated for 3 d, and CD70, which increased arrest in CD8$^+$ T cells pre-activated for 3 or 5 d. Interestingly, CD70—but not CD80 or CD86—is required for dendritic cell-mediated delay of CD8$^+$ T-cell tolerance induction in tumours, confirming a distinct and non-redundant co-stimulatory role for CD70 during early T-cell activation (Bak et al, 2012).

Unlike what we observed in the absence of co-stimulation, these data suggest an intrinsic difference between CD4$^+$ and CD8$^+$ T cells. The fact that CD40 selectively potentiates CD4$^+$ T-cell arrest is consistent with our expression data, which show that CD4$^+$ T cells express approximately three times more CD40L than CD8$^+$ T cells after 3 d of pre-activation. Moreover, CD40L is particularly critical during the licensing of APCs by CD4$^+$ helper T cells, where CD40–CD40L interactions deliver signals that increase the APCs' capacity to prime CD8$^+$ T cells (Gommerman & Summers deLuca, 2011). Beyond licensing, CD4$^+$ T-cell responses are generally considered more dependent on CD40L–CD40 interactions than those of CD8$^+$ T cells (Jones et al, 2000; Iezzi et al, 2009; van Os et al, 2023). Nevertheless, CD40L signalling in T cells remains poorly characterised, and a deeper understanding of how it connects to TCR signalling, calcium flux, and cytoskeletal regulation would be necessary to explain its capacity to potentiate CD4$^+$ T-cell arrest, but not that of CD8$^+$ T cells.

In contrast to CD4$^+$ T cells and CD40L, CD8$^+$ T cells expressed CD27 only ~1.2-fold more than CD4$^+$ T cells, which is probably not sufficient to explain why CD70 selectively promotes the arrest of CD8$^+$ T cells. However, CD70 is critical for CD8$^+$ T-cell activation, after its up-regulation by CD4$^+$ helper T cells through licensing (Laidlaw et al, 2016). Binding of CD27 at the surface of licensed APCs significantly enhances CD8$^+$ T-cell expansion and is often essential for generating robust CD8$^+$ T-cell effector and memory populations (Taraban et al, 2004; Feau et al, 2012), making these cells more reliant on CD27 signalling than CD4$^+$ T cells. In an influenza infection model, the absence of CD27 had only a minor impact on the CD4$^+$ effector pool, whereas the CD8$^+$ T-cell response was severely compromised (Hendriks et al, 2003). Another key feature of CD27, the receptor for CD70, is that it is already expressed at the surface of naïve T cells and can thus regulate the activation of these cells. However, our data show that 3–5 d of activation strongly up-regulates its expression by fourfold. Only these higher expression levels appear to influence CD8$^+$ T-cell arrest, suggesting that CD70 can regulate T-cell activation through two distinct mechanisms: first, by supporting proliferation and survival at low levels of CD27 expression, and second, by promoting T-cell arrest upon pMHC encounter when CD27 is highly expressed.

The precise mechanism by which CD40 and CD70 signalling increases the probability of stable T-cell arrest requires further discussion. By using a Boolean comparison between CHO cells expressing only pMHC and those expressing pMHC plus a single co-stimulatory ligand, we demonstrated that the addition of CD40 or CD70 is sufficient to drive enhanced T-cell arrest. We confirmed the specificity of this system using neutralizing antibodies, which completely reverted the arrest frequency back to baseline levels, demonstrating that direct receptor-ligand engagement is strictly required. Notably, our analysis revealed that the presence of these co-stimulatory molecules increased the overall probability of stable contact formation without accelerating the initial stopping speed of the T cells. We hypothesize that signalling through CD40 or CD27 synergizes with TCR signalling to lower the activation threshold required to commit to a long-term, stable synapse, rather than altering the immediate deceleration kinetics upon antigen encounter. This model is more likely than one in which co-stimulation merely extends the duration of otherwise transient contacts, as we did not observe a corresponding decrease in the frequency of these shorter interactions. While our current study establishes that isolated co-stimulatory pathways are sufficient to modulate T-cell arrest, future studies using primary dendritic cells and multiplexed blockade approaches will be required to define the necessity of these individual pathways within complex, physiological APCs. Dissecting the specific downstream effectors,

such as those regulating the cytoskeleton that would execute this decision to arrest, remains an important direction for future work.

Altogether, the results generated by a novel quantitative "scan and stop" assay show that whereas all T cells can arrest and form an immunological synapse with minimalistic APCs presenting only a cognate pMHC, co-receptors and co-stimulatory molecules modulate this process in a T-cell subset–specific manner. These results reveal a novel aspect of the mechanisms regulating T-cell arrest and, by extension, the role of co-stimulatory molecules in T-cell activation.

Beyond the findings presented here, our "scan and stop" assay provides a versatile platform to address a wide range of questions in T-cell biology. As suggested by our imaging showing APC deformation, the system is well-suited for dissecting the mechanical dimensions of the immune synapse. For example, by engineering the APCs to express an optogenetic actuator like a photo-activatable RhoA protein, the impact of APC stiffness on T-cell arrest and synapse stability could be precisely controlled with light. Moreover, the system's flexibility extends to the surrounding non-activating cell monolayer. These bystander cells could be engineered to recreate aspects of a complex tissue, such as a lymph node or tumour microenvironment. For instance, they could be modified to express specific adhesion molecules or to secrete chemokines and cytokines. This would allow us to quantitatively address how a chemokine-rich environment impacts a T cell's search efficiency, or how the local cytokine milieu modifies the signalling threshold required for stable arrest. Our methodology, therefore, opens up new avenues for a reductionist yet powerful analysis of the many factors governing immunological synapse formation.

# Materials and Methods

## Cell lines

Lenti X293T cells were cultivated in IMDM +10% vol/vol heat-inactivated FCS (iFCS), 1% vol/vol Penicillin+ Streptomycin (Pen/Strep), and 4 mM L-glutamine. PlatE cells (Cell Biolabs) were maintained in PlatE medium (DMEM supplemented with 10% vol/vol iFCS, 1% vol/vol Pen/Strep) for at least 1 wk before viral production. T-Rex CHO cell lines stably expressing single-chain MHC class I (kb-SIINFEKL) co-expressing iRFP713 and T-Rex CHO stably expressing scMHCI loaded with non-cognate VSV-g were a kind gift by Nicolas Gascoigne, The Scripps Research Institute, La Jolla (Hoerter et al, 2013). All CHO cell lines were cultured in Ham's F12 medium supplemented with 10% vol/vol iFCS and 1% vol/vol Pen/Strep at 37°C, 5% $CO_2$. Puromycin (10 $\mu$g/ml) was added to select for co-stimulatory molecule-expressing CHO cell line clones. Used cell lines were tested regularly for mycoplasma contamination.

## Mice

All animals were bred and kept according to institutional, national and European animal regulations and ethics at the central animal facility of the University of Konstanz in appropriate cages with ad libitum access to food and water and a 12/12 light/dark cycle. Male and female mice aged 8–20 wk were used for experiments. Organ collection was approved by the German Veterinary Authority, and animal experiments were authorized by the Review Board of the Regierungspräsidium Freiburg in compliance with the German Animal Protection Law (T-21/03TFA and T-24/02TFA). Transgenic OT-I Mice, expressing a TCR specific for the SIINFEKL (AA 257–264) peptide of ovalbumin, were used to isolate OT-I CD8[+] T cells (Genotype: C57BL/6-Tg [TcraTcrb]1100Mjb). Similarly, transgenic OT-II Mice expressing a TCR specific for the ISQAVHAAHAEINEAGR (AA 323–339) peptide of ovalbumin were used to isolate OT-II CD4[+] T cells (Genotype: B6.Cg-Tg[TcraTcrb]425Cbn/J). C57BL/6J mice were used to isolate and cultivate BMDCs.

## T-cell isolation and culture

Mice were euthanized by $CO_2$, and spleens were harvested in sterile PBS (PAN Biotech). Spleens were mechanically dissociated through a 70-$\mu$m strainer using a syringe plunger and washed with cold FACS buffer (PBS +2% FCS + 2 mM EDTA). After centrifugation (300$g$, 10 min, 4°C) and RBC lysis (5 ml lysis buffer, 1 min, RT; Lucerna), cells were counted. Naïve CD8[+] or CD4[+] T cells were enriched using the mouse CD4[+] or CD8$\alpha$[+] naïve T-cell isolation kit (Miltenyi Biotech) by means of MACS following the manufacturer's protocol. The T cells were afterwards resuspended in culture medium (PAN Biotech RPMI 1640 + 10% vol/vol iFCS, 1% vol/vol Pen/Strep, 50 $\mu$M $\beta$-mercaptoethanol) at 1 × 10^6 cells/ml and seeded into six-well plates (Sarstedt). To generate antigen-experienced T-cell populations for the scan and stop assay, activation of naïve T cells was performed using anti-CD3/CD28 Dynabeads (1:1 bead: cell ratio; Thermo Fisher Scientific) with 40 U/ml murine IL-7 (Peprotech); the following day, human IL-2 was added (100 U/ml; Peprotech). Cultures were maintained at 37°C with medium refreshed as needed.

On day 4 after activation, density gradient centrifugation was performed using Ficoll-Paque (Cytiva) to remove dead cells. Live cells were collected from the interphase above the Ficoll layer and transferred into a fresh tube containing 10 ml of culture medium. Cells were washed twice with fresh medium. Before the final centrifugation, cells were counted and seeded at 1 × 10^6 cells/ml in fresh culture medium supplemented with 100 U/ml IL-2 and 40 U/ml IL-7 and incubated overnight at 37°C to allow recovery.

## BMDC isolation

BMDCs from WT C57BL/6J mice were generated as previously described (Lutz et al, 1999). In brief, BM cells were harvested by flushing femurs and tibiae with PBS using a syringe, followed by centrifugation at 300$g$ for 5 min. RBCs were lysed using 1× RBC lysis buffer for 30 s at RT. The cell suspension was passed through a 70 $\mu$m cell strainer to remove debris such as bone fragments and neutralized with PBS. After centrifugation, the cells were resuspended in R10 culture medium (RPMI 1640 supplemented with 2 mM L-glutamine, 10% vol/vol iFCS, 1% vol/vol Pen/Strep and 50 $\mu$M $\beta$-mercaptoethanol). The cells were seeded at 4 × 10^6 cells/ml in bacteriological 10-cm Petri dishes containing 10 ml of R10 culture medium supplemented with 20 ng/ml murine

granulocyte-macrophage colony-stimulating factor (GM-CSF; Peprotech). After 3 d of incubation at 37°C in a humid atmosphere with 5% $CO_2$, 10 ml of R10 medium containing 20 ng/ml GM-CSF was added to the cultures. Half of the culture medium was replaced 3 d later by carefully aspirating media from the top and replenishing it with fresh R10 medium containing 20 ng/ml GM-CSF. Cells were considered as immature BMDCs on days 8 and 9 of in vitro differentiation. Maturation of BMDCs was induced on day 8 or 9 by resuspending the cells in R10 medium containing GM-CSF and 100 ng/ml lipopolysaccharide from *Escherichia coli* O111:B4 for 20 h (LPS; Sigma-Aldrich).

### Plasmids

For the generation of CHO cell lines and the production of retroviral particles, a range of plasmids was used. Plasmids encoding single-chain MHC class II (H2-I-A(b)-ISQAVHAAHAEINEAGR) were synthesized by GeneScript (2333; Leiden) with subsequent haplotype exchange. A detailed protocol is included in the supplementary material. To express co-stimulatory ligands, we used plasmids encoding mouse receptors, which were either obtained from Addgene or OriGene. 4-1BB-L (#MR220933L4; OriGene), CD40 (#121166; Addgene), HVEM (#MR223946L4; OriGene), OX40L (#MR223452L4; OriGene), and CD70 (#MR225924; OriGene). Murine CD80 (#MR227446L4; Ori-Gene). For each construct, the monomeric GFP tag was cloned to the intracellular part of the molecule using Gibson assembly where needed. The retroviral packaging plasmids pECO and VSV-G, used for virus production in HEK293T cells, were obtained from Clonetech, Cat# 631530. The retroviral construct used to transduce T cells with mCherry-tagged CD40L was generated by Gibson assembly, cloning CD40L (#121167; Addgene) into a pMSCV-mScarlet backbone (#21654; Addgene). All plasmids were sequence-verified before use.

### Lentivirus production

For producing pseudotyped lentiviruses, Lenti-X 293T cells (Takara Bio Inc.) grown in poly-L-lysine–coated vessels (100 $\mu$g/ml, 30 min at 37°C; Sigma-Aldrich) and antibiotic-free medium were transfected with (I) the envelope plasmid pMDG.2 encoding for the VSV-G glycoprotein, (II) the packaging plasmid psPAX2 encoding structural and regulatory proteins, as well as (III) a transfer plasmid containing the co-stimulatory molecule of interest. Thereby, plasmid DNA was diluted in OptiMEM (Life Technologies Corporation), in the ratio of 1:1.8:2 (pMD2.G: psPAX2: expression vector). MIRUS LT1 transfection reagent (Mirus Bio LLC) was added at a 3:1 reagent-to-DNA ratio, immediately mixed, and incubated at room temperature for 20–30 min. The transfection mixture was then added dropwise to the cells and the medium was changed 8 h post-transfection to minimize cytotoxic effects of VSV-G envelope expression. Viral supernatants were collected 36–48 h post-transfection (with subsequent collections every 12–24 h up to 72 h), filtered (0.45 $\mu$m), and treated with DNase (1 $\mu$g/ml with 1 mM $MgCl_2$, 20 min at 37°C; Roche Diagnostics). Supernatants were either concentrated or aliquoted and frozen at –80°C. Lentiviral particle concentration was achieved by PEG 6000 precipitation (8.5% PEG + 0.28 M NaCl). In this regard, particles were incubated overnight with PEG at 4°C, followed by centrifugation at 4,650$g$ for 32 min the next day. The resulting pellet was carefully resuspended in PBS, aliquoted, and stored at –80°C.

### Titration of lentiviral stocks by flow cytometry

Lenti-X 293T cells were seeded at 0.5–1 × $10^5$ cells/well in 12-well plates. Before transduction, cells from two wells were counted, and the recorded number was retained for later use in titre calculation. Serial ten-fold dilutions ($10^0$–$10^{-5}$) of viral stock were added along with a mock control. After 3–4 d at 37°C, cells were detached with trypsin/EDTA, washed with FACS buffer, and analysed by flow cytometry. Titre (TU/ml) was calculated using:

$$Titre\left(\frac{transducing\ units}{ml}\right) = \frac{number\ of\ cells\ before\ transduction \times \frac{\%reporter\ positive\ cells}{100}}{volume\ of\ virus\ added}cells\ (ml)$$

For accurate titre calculation, only dilutions resulting in 1–30% reporter-positive cells were considered.

### Transduction of CHO cells for co-stimulatory ligand expression

CHO cells (65–80% confluency) had their medium changed to puromycin-free medium 1 h pre-transduction. Thereafter, protamine sulphate (1:100; Lucerna) was added to prevent repulsion between cells and viral particles. Viral supernatant was applied dropwise (MOI 50–100) to CHO cells. Cells were incubated at 37°C for ≥24 h and subsequently tested in flow cytometry for the expression of co-stimulatory molecules.

### Transduction of T cells

The transduction of T-cell protocol was adapted from the study by reference Eremenko et al (2021) containing the following workflow: 80–90% confluent PlatE cells (in a poly-L-lysine–coated vessel) were transfected with the retroviral construct of interest using Mirus LTI, as described above. After 8 h, the medium was replaced with 9 ml of PlatE medium containing 0.1% vol/vol Pen/Strep. Viral supernatants were harvested 48–60 h post-transfection and filtered (0.45 $\mu$m) for subsequent use. On the day after transfection, splenocytes from OT-II mice were isolated, and CD4$^+$ T cells were enriched, as described above. These cells were cultured in RPMI supplemented with 10% vol/vol iFCS, 1% vol/vol Pen/Strep, and 50 $\mu$M $\beta$-mercaptoethanol, and activated with Dynabeads at a 1:1 bead-to-cell ratio for 18–24 h. Simultaneously, a non-treated 24-well plate was coated overnight at 4°C with PBS containing 20 $\mu$g/ml RetroNectin (Takara Bio Inc.). On day 2 post-transfection, the coated wells were blocked with 500 $\mu$l/well PBS containing 2% BSA for 30 min at RT, followed by two PBS washes. Meanwhile, activated T cells were supplemented with 100 U/ml hIL-2 for 2 h. The filtered viral supernatant was added (1 ml per well) to the RetroNectin-coated plate, and the plate was centrifuged at 2,000$g$ for 2 h at 32°C. Shortly before the centrifugation ended, T cells were collected, centrifuged at 300$g$ for 10 min, and resuspended in the retained supernatant to 1 × $10^6$ cells/ml. Dynabeads were removed magnetically, and 10 mM Hepes (Sigma-Aldrich) was added. After the removal of residual viral supernatant from the plate, 1 × $10^6$

T cells were added per well and further centrifuged at 800$g$ for 30 min at 32°C before incubation at 37°C, 5% $CO_2$. After 48 h, half of the culture medium was replaced with fresh T-cell medium supplemented with 20 U/ml IL-2 and 10 ng/ml IL-7.

### Scan and stop assay

CHO-OVA1 target cells (T-Rex CHO cells expressing scKb–OVA and p2A iRFP713) and CHO-VSV non-target control cells (T-Rex CHO cells expressing scKb-VSV under an hCMV promoter) were used. Alternatively, CHO-OVA2 cells expressing sclAb-ISQAVHAAHAEINEAGR and parental CHO WT cells served as the target and control groups, respectively. Imaging coverslips (thickness 1.5, 18 mm) were coated in 12-well plates with fibronectin (10 $\mu$g/ml) for at least 30 min at 37°C. Cells were harvested, resuspended in complete medium, and cell densities adjusted for seeding into different chamber slides. Target and control CHO cells were mixed at a ratio of 1:15 and allowed to adhere for 20 min before removal of non-adherent cells. In case of blocking CD40 (Cat.# 102802; BioLegend) or CD70 (Cat.#: 16-0701-82; Invitrogen) on the CHO cells, the respective antibody (10 $\mu$g/ml) was added 20 min before imaging and kept on the coverslip during imaging. T cells were labelled with CellTracker Blue (1:50 dilution in PBS, 30 min at 37°C; Thermo Fisher Scientific), adjusted to 1 × 10$^6$ cells/ml (with IL-7 and IL-2, if needed), and added to the CHO monolayer in imaging chambers containing 300 $\mu$l medium. After a 15-min settling period, epifluorescence imaging was performed on a Leica DMi8 TIRF microscope using a 40× air objective and a mercury lamp, with tile scans (3 × 3, 10% overlap) acquired at 1 frame/min for 91 frames. Imaging was performed fully inside an incubation chamber, heated to 37°C, 5% $CO_2$. Images were processed in LAS-X (merged, resized to 50%, converted to 8-bit) and further analysed using the latest version of FIJI/ImageJ.

### Scan and stop analysis

The multi-channel time-lapse image files (.lif) were loaded into ImageJ, and single channels of each image were split and saved as .tiff, respectively.

### Analysis of target cells

After loading the respective file into ImageJ, brightness and contrast were manually optimized to ensure the best separation between CHO cells and background noise. A threshold was applied using the Otsu pre-set method. The settings were manually adjusted for optimal target cell detection. A Gaussian blur filter (radius = 2) was applied, followed by binary conversion. If necessary, touching target cells were manually separated using the freehand tool. Pixels to be removed were selected and deleted in all necessary frames.

### Analysis of T cells

Conversion into a binary picture was performed as described above. For naïve T cells, the watershed function was applied to separate touching cells.

### Cell tracking

T cells and CHO cells were tracked using the TrackMate plugin of ImageJ, using the Mask detector function. A filter by area was applied individually to remove small artifacts and background noise. Cells were tracked using the Simple LAP tracker. Manual spot addition and linkage were performed as needed to refine tracking accuracy. The tracking was manually verified against raw images to ensure accuracy in movement detection.

### Track analysis using the script

The corresponding -t.xml and -target.xml files were opened in the purposely programmed HTML GUI for analysis. Within the HTML GUI, we applied upscaling and filtering of T-cell tracks to improve data quality and reduce noise. T-cell polygons were upscaled by 0.2% to account for binary signal loss at the borders of the T cells. Afterwards, we filtered for tracks with Euclidean distances of less than 15 $\mu$m to exclude non-moving or dead T cells. Within the GUI, the parameters for contacts were defined as well: A "touch" was defined as any non-empty intersection between a T-cell polygon and a CHO cell polygon, provided that the intersection area covered at least 10% of the T-cell surface. Consecutive touches were grouped into "contacts," including contacts consisting of a single frame. To account for brief fluctuations in cell segmentation or unstable edge contacts, interruptions of up to five consecutive frames without detectable overlap were tolerated within a contact sequence. Contacts that persisted until the end of the 90 min acquisition period were classified as ""stable contacts." The contact information was exported as an Excel file. Tracks with a persistence of more than 0.5 were not considered for the analysis, as they stem from drifting, not adherend T cells. Details on the contact analysis routine can be found under the GitHub link provided in the Data Availability section.

### Flow cytometry

Cells were centrifuged at 300$g$ for 5 min, and the supernatant was carefully removed. Cells were washed twice with FACS buffer, leaving ~100 $\mu$l of residual buffer after the final wash. Antibody staining was performed by adding the respective antibody at a 1: 50 dilution in an additional 100 $\mu$l of FACS buffer, resulting in a final staining volume of 200 $\mu$l, if not stated otherwise. Co-stimulatory molecule expression on CD4$^+$ and CD8$^+$ T cells was detected using the following antibodies against: TCR-V$\alpha$2 SB780 (Cat.#: 78-5812-82; Invitrogen), CD27 APC (Cat.#: 124211; BioLegend), CD134 (OX40) APC (Cat.#: 119413; BioLegend), CD137 (4-1BB) APC (Cat.#: 106109; BioLegend), CD154 (CD40L) APC (Cat.#: 106510; 1:50; BioLegend). T-cell subsets of CD4$^+$ and CD8$^+$ T cells (naïve: CD44$^-$ CD62L+; effector/ effector memory: CD44$^+$ CD62L–; central memory: CD44$^+$ CD62L+) were determined using the following antibodies against: TCR-V$\alpha$2 SB780 (Cat.#: 78-5812-82; Invitrogen), CD44 AF488 (Cat.#: 103015; BioLegend) and CD62L PE (Cat.#: 104408; BioLegend). Transduced CHO cell lines were confirmed via flow cytometry using the following antibodies: CD80 (Cat. #: 104715; BioLegend), 4-1BB-L (Cat.#: 50067R014P; SinoBiological), CD40 (Cat.#: 12-0401-82; Invitrogen), OX40L (Cat.#: 108811; BioLegend), CD70 (Cat.#: 104606; BioLegend),

HVEM (Cat.#: ab47677; Abcam with Cat.#: a21246; Invitrogen as secondary) and MHC-I (Cat.#: 116525; BioLegend). BMDCs were stained for CD11c (Cat.#: 117310; BioLegend), MHC-II (Cat.#: 107632; BioLegend), and MHC-I (Cat.#: 116525; BioLegend). All analyses were exclusively conducted on the CD11c[+] MHC-II[+] population to ensure that only fully differentiated cells were included. Co-stimulatory molecules on mBMDCs were measured with the same antibodies as the CHO cell lines. Appropriate isotype controls were used. Samples were incubated for 20 min at 4°C in the dark. After incubation, cells were washed twice with FACS buffer and stored at 4°C until flow cytometry analysis.

### scMHCII cloning

#### Insertion of the correct alpha chain of H2-I-A b haplotype

**Design primers for overlap extension PCR:** One part of each primer binds to the alpha chain of the H2-I-A b haplotype cDNA generated from total.

RNA isolated from WT BMDCs that were stimulated with 100 ng/ml LPS and 600 U/ml IFN-g for 24 h. The other part of each primer binds to regions downstream and upstream of the wrong alpha chain of the H2-I-A d haplotype. In the first PCR, the "Mega"-primer is generated that is used to insert the correct alpha chain into the pcDNA4/TO-sc-MHCII-OTII by overlap-extension PCR.

These long primers are used to amplify the alpha chain from BMDC cDNA, resulting in a 690-bp long amplicon, the "Mega"-primer. Bold and underlined sequences are present in both the plasmid pcDNA4/TO-scMHCII-OTII and the cDNA from BMDCs.

The amplified fragment/insert is used as a "Mega"-primer in the overlap extension PCR.

Sequence of amplified insert/"Mega"-primer pcDNA4-TO-IAb-alpha chain:

5'-TAAACTTAAGCTTGGTACCGCCACCATGCCGCGCAGCAGAGCTCTG ATTCTGGGGGTCCTCGCCCTGACCACCATGCTCAGCCTCTGTGGAGGTGAA GACGACATTGAGGCCGACCACGTAGGCACCTATGGTATAAGTGTATATCAG TCTCCTGGAGACATTGGCCAGTACACATTTGAATTTGATGGTGATGAGTTG TTCTATGTGGACTTGGATAAGAAGGAGACTGTCTGGATGCTTCCTGAGTTT GGCCAATTGGCAAGCTTTGACCCCCAAGGTGGACTGCAAAACATAGCTGTA GTAAAACACAACTTGGGAGTCTTGACTAAGAGGTCAAATTCCACCCCAGCT ACCAATGAGGCTCCTCAAGCGACTGTGTTCCCCAAGTCCCCTGTGCTGCTG GGTCAGCCCAACACCCTCATCTGCTTTGTGGACAACATCTTCCCTCCTGTG ATCAACATCACATGGCTCAGAAATAGCAAGTCAGTCGCAGACGGTGTTTAT GAGACCAGCTTCTTCGTCAACCGTGACTATTCCTTCCACAAGCTGTCTTAT CTCACCTTCATCCCTTCTGACGATGACATTTATGACTGCAAGGTGGAACAC TGGGGCCTGGAGGAGCCGGTTCTGAAACACTGGGAACCTGAGATTCC AGC CCCCATGTCAGAGCTGACCGAGACAGGCGGCGG-3'

Orange = binding in pcDNA4/TO-scMHCII-OTII.
Blue = binding in H2-IA beta alpha chain.
Black = binding in both.
Green = amplified alpha chain of beta haplotype.

**Conventional PCR primer:**

| Name | Sequence (5'-3') | Tm/Ta (Tm Calculator Thermo Fisher Scientific) | | |
|---|---|---|---|---|
| H2-IAb alpha chain_for | ATGCCGCGCAGCAGAGCTCTGATTC | 70.9°C | 72°C | This annealing temperature is used for amplifying the correct alpha chain from BMDC cDNA |
| H2-IAb alpha chain_rev | GTCAGCTCTGACATGGGGGCTGGAATCTC | 71.1°C | | |
| pcDNA4-TO_rev | CCGCCGCCTGTCTCGGTCAG | 69.1°C | 72°C | This annealing temperature is used for the overlap |
| pcDNA4-TO_for | TAAACTTAAGCTTGGTACCGCCACCATGCC | 69.9°C | | extension PCR used to insert the correct alpha chain into the plasmid pcDNA4/TOscMHCII-OTII |

**Overlap extension PCR primer (2 combined conventional PCR primers plus additional 65 restriction enzyme sites):**

| Name | Sequence (5'-3') |
|---|---|
| Overlap-I-Ab alpha chain_for | TAAACTTAAGCTTGGTACCGCCACC**ATGCC**GCGCAGCAGAGCTCTGATTC |
| Overlap-I-Ab alpha chain_rev | CCGCCGCCTGTCTCG**GTCAG**CTCTGACATGGGGGCTGGAATCTC |

**Amplification of "Mega"-primer: Reaction setup to create "Mega"-primer from BMDC cDNA**

| Reagent | Volume | Final Conc. |
|---|---|---|
| 5x Phusion HF buffer | 10 µl | 1× |
| cDNA template (50 ng/µl) (up to 10% of final volume) | 2 µl | 100 ng |
| 10 µM forward primer | 2.5 µl | 0.5 µM |
| 10 µM reverse primer | 2.5 µl | 0.5 µM |
| 100% DMSO | 1.5 µl | 3% |
| 10 mM dNTP-Mix | 1 µl | 200 µM |
| 2 U/µl Phusion polymerase | 0.5 µl | 0.02 U/µL |
| ddH2O/MQ | Add to 50 µl (30 µl) | |

**PCR cycling conditions (conventional PCR):**

| Step | Temperature | Time | Cycles |
|------|-------------|------|--------|
| Initial Denaturation | 98°C | 30 sec | 1 |
| Denaturation | 98°C | 20 sec | 30 cycles |
| Annealing/Elongation | 72°C | 28 sec (40 sec/kb, 0.690 kb) | |
| Final Elongation | 72°C | 10 min | 1 |
| Cooling | 4°C | Hold | 1 |

**Overlap-extension PCR: Reaction setup for overlap-extension PCR using the "Mega"-primer and pcDNA4/TO-scMHCIIOTII: (molar ratio 1:250 of vector template: "Mega"-primer).**

| Reagent | Volume | Final Conc. |
|---------|--------|-------------|
| 5x Phusion HF buffer | 10 µl | 1x |
| Plasmid template(stock 500 ng/µl) | 2 µl (of 1:100 dilution of stock 500 ng/µl) | 10 ng |
| Insert/"Mega"-primer (63 ng/µl) | 5 µl | 315 ng (~302 molar excess) |
| 100% DMSO | 1.5 µl | 3% |
| 10 mM dNTP-Mix | 1 µl | 200 µM |
| 2 U/µl Phusion polymerase | 0.5 µl | 0.02 U/µl |
| ddH2O/MQ | Add to 50 µl | |

**PCR cycling conditions (overlap-extension PCR):**

| Step | Temperature | Time | Cycles |
|------|-------------|------|--------|
| Initial Denaturation | 98°C | 30 sec | 1 |
| Denaturation | 98°C | 20 sec | 20 cycles |
| Annealing/Elongation | 72°C | 210 sec (30 sec/kb, 6.630 kb) | |
| Final Elongation | 72°C | 10 min | 1 |
| Cooling | 4°C | Hold | 1 |

Underlined = complete correct alpha chain sequence (until T219).

During the PCR, the alpha chain of haplotype D should be exchanged with the correct alpha chain of haplotype B. The plasmid should be exactly the same size of 6,630 bp.

After the PCR, a *DpnI* digest was performed (1 µl added directly to the reaction and incubated for 1.5 h at 37°C, then heat-inactivated at 80°C for 10 min). 10 µl of the PCR reaction was transformed directly into 100 µl competent *E.coli* NEB stable, and the entire mix (spun down at 6,700*g*, 2 min, RT) was spread on an LB plate containing 100 µg/µl ampicillin. The plates were incubated o/n at 37°C. The next day, a few colonies had grown. 10 colonies were picked and inoculated in 5 ml LB cultures with 100 µg/ml ampicillin. They were grown o/n at 37°C, and then sent for sequencing.

## Sequence for scMHC-II

GCCACCATGCCCTGCAGCAGAGCCCTGATCCTGGGCGTGCTGGCTCTGAAC
ACCATGCTGTCCCTGTGCGGCGGCGAGGATGTGATTGAGGCTGACCACGTG
GGCTTTTACGGCACCACAGTGTATCAGTCCCCTGGCGATATCGGCCAGTAC
ACCCACGAGTTCGATGGCGACGAGCTGTTTTATGTGGATCTGGACAAGAAG

AAGACCGTGTGGAGGCTGCCCAGAGTTCGGCCAGCTGATCCTGTTTGAACCT
CAGGGCGGCCTGCAGAACATTGCTGCTGAGAAGCACAATCTGGGCATCCTG
ACCAAGAGATCTAACTTTACCCCAGCTACAAATGAGGCCCCTCAGGCTACC
GTGTTCCCTAAGAGCCCAGTGCTGCTGGGCCAGCCTCACACACTGATCTGC
TTCGTGGACAACATCTTTCCCCCTGTGATCAACATCACCTGGCTGCGGAACA
GCAAGTCTGTGACCGATGGCGTGTACGAGACATCCTTCCTGGTGAATCGGGAC
CACTCCTTTCACAAGCTGTCTTACCTGACATTCATCCCATCTGACGATGACAT
CTATGATTGTAAGGTGGAGCACTGGGGCCTGGAGGAGCCTGTGCTGAAG
CACTGGGAGCCAGAGATCCCAGCCCCCATGTCTGGCCTGACCGAGACAGGC
GGCGGCGGCAGCGGCGGCGGCGGCAGCCAGCAGGGCAGGCTGGACAAG
CTGACCATCACATCCCAGAACCTGCAGCTGGAGTCTCTGAGAATGAAGCTG
CCCAAGTCCGCCATCTCTCAGGCTGTGCACGCCGCTCACGCCGAGATCAAT
GAGGCTGGCAGGGGCGGCGGCTCTAGCGGCGGCGGCGGCAGCGGCGGC
GGCGGCTCTGAGCGGCACTTCGTGTACCAGTTTATGGGCGAGTGCTACTTC
ACCAACGGCACACAGCGGATCAGGTACGTGACCCGGTACATCTATAATAGG
GAGGAGTACGTGAGATATGATTCTGACGTGGGAGAGCACAGAGCTGTGACA
GAGCTGGGCAGGCCCGATGCTGAGTACTGGAACAGCCAGCCTGAGATCCTG
GAGAGAACAAGAGCTGAGCTGGACACAGTGTGCAGACACAACTATGAGGGC
CCTGAGACACACACATCTCTGCGGAGGCTGGAGCAGCCAAACGTGGTCATC
TCTCTGTCCAGAACAGAGGCCCTGAACCACCACAATACCCTGGTGTGCAGCGTG
ACAGACTTCTACCCAGCTAAGATCAAGGTGAGATGGTTCCGCAACGGCCAG

GAGGAGACAGTGGGCGTGTCTAGCACACAGCTGATCCGGAATGGCGACTGGACC
TTCCAGGTGCTGGTCATGCTGGAGATGACCCCCAGACGCGGCGAGGTGTAT
ACATGCCACGTGGAGCACCCCAGCCTGAAGTCCCCTATCACAGTGGAATGG
AGAGCTCAGTCTGAGAGCGCCTGGTCTAAGATGCTGTCTGGAATTGGCGGA
TGCGTGCTGGGCGTGATCTTCCTGGGCCTGGGCCTGTTTATCCGGCACAGG
AGCCAGAAGGGCCCTCGGGGCCCACCCCCTGCCGGCCTGCTGCAGTGAGAA
TTC.

This sequence was cloned into Invitrogen #V102020 and subsequently used as a template in this protocol. The resulting insert was subsequently cloned into a pMSCV backbone provided by Szun Tay, UNSW Sydney, via restriction digest.

# Data Availability

The raw data supporting the findings of this study have been deposited on Zenodo and are publicly available under the https://doi.org/10.5281/zenodo.15275955. The dataset includes ImageJ tracking files, the corresponding Excel-based analysis, and flow cytometry data. Because of file size limitations, the original live-cell imaging data used for tracking are not included in the repository but are available from the corresponding author upon reasonable request. The source code for the HTML-based analysis routine can be accessed at: https://github.com/dbuenzli/cell-contacts.

# Supplementary Information

# Acknowledgements

We would like to sincerely thank Daniel Bünzli for his invaluable support in coding and providing the HTML-based analysis routine used in this study. We are also grateful to Michael Basler (University of Constance, Konstanz, Germany) for his dedicated efforts in managing all regulatory aspects related to the animal facility at the University of Konstanz. In addition, we thank Camille Clamagirand (Institute of Cellular Biology and Immunology Thurgau, Kreuzlingen, Switzerland) for generously providing bone marrow–derived dendritic cells (BMDCs), which contributed critically to the experimental setup. The authors thank the animal facility of the University of Konstanz for their support in animal husbandry and experimental procedures. We also acknowledge the support of the Thurgauische Stiftung für Wissenschaft und Forschung. Furthermore, we are grateful to the Graduate School for Cellular and Biomedical Sciences (GCB) at the University of Bern for providing an excellent training environment and academic support. This study was funded by grants from the Deutsche Forschungsgemeinschaft (RO6238/1-1 and RO6238/2-1 to J Rossy) and the Swiss Science Foundation (310030_201059 and 31003A_172969 to J Rossy) and the State Secretariat for Education, Research and Innovation.

## Author Contributions

V Gloe: data curation, formal analysis, investigation, visualization, methodology, and writing—original draft.

R Schregle: investigation, methodology, and writing—review and editing.

C Ratswohl: formal analysis, investigation, methodology, and writing—review and editing.

J Rossy: conceptualization, supervision, project administration, and writing—original draft, review, and editing.

## Conflict of Interest Statement

The authors declare that they have no conflict of interest.

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
