## [Reviewer comments · Life Science Alliance]

A 'scan and stop' assay identifies CD40 and CD70 as selective regulators of T cell arrest on APCs

Vincent Gloe, Richard Schregle, Christoph Ratswohl, and Jérémie Rossy

DOI: <https://doi.org/10.26508/lsa.202503401>

Corresponding author(s): Jérémie Rossy, Institute of Cell Biology and Immunology Thurgau

Review Timeline:

Submission Date:	2025-05-27
Editorial Decision:	2025-07-23
Revision Received:	2026-04-14
Editorial Decision:	2026-05-19
Revision Received:	2026-05-22
Accepted:	2026-05-26

Scientific Editor: Tim Fessenden

Transaction Report:

July 23, 2025

Re: Life Science Alliance manuscript #LSA-2025-03401

Dr. Jérémie Rossy
Institute of Cell Biology and Immunology Thurgau
Unterseestraße 47
Kreuzlingen s 8280
Switzerland

Dear Dr. Rossy,

Thank you for submitting your manuscript entitled "A 'scan and stop' assay identifies CD40 and CD70 as selective regulators of T cell arrest on APCs" to Life Science Alliance. The manuscript was assessed by expert reviewers, whose comments are appended to this letter.

As you will see, all reviewers appreciated the observations on receptor-ligand engagement capable of modulating T cell arrest on target cells. However, reviewers remarked on validations needed to fully support the claims made, in particular related to expression levels of receptors involved and on levels of Ova peptides presented on target cells (Reviewer 1 points 2 and 5, Reviewer 3 points 1 and 2). The use of blocking antibodies was requested to confirm the arrest requires receptor-ligand interactions (Reviewer 1 point 6), which may also be leveraged to derive greater mechanistic insight as suggested by Reviewer 2. A revision should address all remaining points by Reviewer 3, including quantification requested in point 4, however additional data not noted here are not required in a revised manuscript.

Thank you for this interesting contribution to Life Science Alliance. We are looking forward to receiving your revised manuscript.

Sincerely,

-- Summary blurb (enter in submission system): A short text summarizing in a single sentence the study (max. 200 characters including spaces). This text is used in conjunction with the titles of papers, hence should be informative and complementary to

the title and running title. It should describe the context and significance of the findings for a general readership; it should be written in the present tense and refer to the work in the third person. Author names should not be mentioned.

B. MANUSCRIPT ORGANIZATION AND FORMATTING:

Reviewer #1 (Comments to the Authors (Required)):

This is an intriguing study revealing that costimulatory ligands CD40 and CD70 promote CD4 and CD8 T cell arrest on APCs, respectively. The study was based on coculturing OT-II or OT-I T cells with artificial antigen presenting cells: CHO cells overexpressing a single chain pMHC (for TCR activation) and either CD40 or CD70. While the conceptual message is interesting, the study can be strengthened in multiple aspect in order to be more complete and convincing. I have the following suggestions to improve the study.

Major:

1. If the T cell arrest occurs at minutes time scale, what is the rationale of measuring the effects of pre-activation and/or costimulatory ligands on different days? Wouldn't it be more informative to measure these effects over the time course of minutes or hours?
2. To improve the physiological relevance of the findings, can the authors please measure the expression of the costimulatory ligands on the CHO cells using flow cytometry and compare these with those on dendritic cells?
3. What is the rationale of using different set of ligands for CD4 (Fig. 3) and CD8 (Fig. 4) T cells? HVEM and OX40L can be included for CD4 experiments to make it consistent with the CD8 conditions.
4. Some data in Fig. 3 and Fig. 4 lack day 3 conditions. Can the authors please add the condition? Indeed, CD40 significantly increases the % stable contact on day 3. This could be true for other ligands too.
5. Fig. 5, CD40L-mCherry appears largely intracellular, is it normal or an artifact introduced by the mCherry tag? Is this possible to use a different fluorescent tag or immunostain the endogenous CD40L? Similar experiments can be done for CD70 and its receptor CD27 to make this study more complete.
6. The study can be benefited from testing whether the addition of blockade antibodies that disrupt either CD40-CD40L or CD70-CD27 interactions abrogate the effects observed.
7. Related to the point above, the antibody blockade approach could allow the authors to test using peptide loaded DCs as a more physiological APCs, and test the necessity of the CD40-CD40L or CD70-CD27 interactions in promoting the T cell stop. Their minimalistic CHO system is a nice way to test the sufficiency, but not the necessity.

Minor:

1. Color scheme of the bar graphs a bit confusing. For example, in Fig. 4B-G, the color of some of the bars are similar to the color of the molecules/cells in Fig. 4A. I suggest that authors use open bars to indicate pMHC-I alone condition and the filled bar to indicate +costim ligand conditions.
2. "We first transduced pMHC-expressing CHO cells with CD80, which is the ligand for CD28". I'd change "the ligand" to "a ligand" because CD80 is not the only ligand for CD28.
3. The rationale of testing CD70 was not clearly described.

Reviewer #2 (Comments to the Authors (Required)):

The manuscript titled "'scan and stop' assay identifies CD40 and CD70 as selective regulators of T cell arrest on APCs" by Gloe et al. utilizes a well-established property of arresting when encountering an agonist antigen to identify co-stimulatory signals required to achieve an arrest. Indeed, through a number of intravital imaging studies, the arrest of T cells (or "stop signal") has been established as a hallmark of T cell activation. Gloe et al. leverage this property nicely to generate a controlled ex vivo assay for identifying co-stimulatory regulators of arrest. For this, develop a cellular antigen presentation system that has minimal background engagements, thereby revealing quantitative information on phenotype resulting from the co-stimulatory signal alone. The assay system is neat and provides a crucial testing ground for the role of surface receptors in a more physiologically

relevant mechanical context than possible with most planar ligand reconstitution systems that are conventionally used for such assays. Overall, their system is clean, experiments are well constructed and presented with appropriate controls, and the manuscript is also written very well. I highly recommend it for the publication.

My only suggestions are:

1. The authors identify CD70 as a surprising ligand inducing arrest. They have nicely discussed the known literature on CD70. However, more discussion on what it may be doing is missing. This is best discussed, or at least speculated, in the context of synapse formation and breaking kinetics. For instance, does CD70 promote entry into arrest, or prolong the arrest phase (e.g., Kumari et al. EMBO J., 2020), or both? Perhaps authors could add CD70 neutralizing antibody after initial arrest and examine if the arrest is broken, and T cells start migrating again to address this?
2. Following up on the previous point, the authors' assay system is uniquely suited to address the many mechanical dimensions of receptor function at the synaptic interface as well as overall synapse evolution and lifetime. In addition, the manuscript dedicates great effort to setting up and characterizing the system itself. It will be great if authors can devote a paragraph in the discussion to highlight the broad questions in T cell biology that their methodology can address in the future.

Reviewer #3 (Comments to the Authors (Required)):

In this manuscript, Gloe et al. designed an in vitro T-APC assay (stop and scan) that leverages CHO cells engineered to express pMHC complex as well as selected co-stimulatory molecules to assess the role of these molecules in the arrest of antigen-specific T cells with their antigen-presenting cells. While not particularly novel, this method provides a useful platform for reductionistic evaluation of molecules that control T-APC interactions. Using this approach, the authors identified CD40L on CD4+ T cells and CD27 on CD8+ T cells - both known co-stimulatory molecules - as having potentially novel roles in mediating T cell arrest during priming with APCs.

The manuscript is well written, and the experiments were, for the most part, well controlled. The discussion section also addressed the limitations and some of the questions that this reviewer had while reading through the manuscript.

Specific points that need to be addressed:

1. In Fig. 1D-E and Fig. 2, the authors showed that the formation of "stable contact" as well as the arrest parameters were not different between naïve and Day 3 and Day 5 pre-activated T cells. Were the TCR/CD3 expression level of the pre-activated T cells different (e.g. downregulated) compared to naïve T cells?

Downregulation of TCRs during activation can influence binding avidity with pMHC and consequently the T cell arrest behavior with APC, and the TCR levels should at least be measured to address whether the avidity affects T-APC interaction.

2. Related to the above point, it is unclear if the transduced levels of single-chain OVAp-MHC in CHO cells are within physiological range or if excess amounts of pMHC signals were being presented to the antigen-specific T cells coming into contact.

Although the authors showed that the detected level of MHC-I was similar between BMDCs and CHO cells (Supp. Fig. 1G), note that unless the OVA-pMHC level was measured (e.g. using 25D1 antibody for SIINFEKL and similar OVA 323-339 antibody for pMHC-II), it is difficult to assess if the antigen level being presented is within physiological range. We may assume that all of the mouse MHC molecules detected on CHO cells were the transduced single-chain OVA peptide-MHC complexes, and if these cells express similar levels of MHC-I and MHC-II as BMDCs, it would amount to that all of the MHC molecules expressed by BMDCs were presenting the OVA antigens.

As such, excessive amount of antigen could overcome TCR downregulation (see point #1 above) and the drop in TCR avidity, and therefore obfuscates the T cell arrest dynamics between naïve and activated T cells that could have been different under physiological contexts.

The authors can control for this by staining for the OVA peptide-MHC complexes on the engineered CHO cells and compare them to those measured on BMDCs incubated with a reasonable amount of OVA antigen where the presented antigens have gone through antigen processing steps under physiological conditions, or at least address this limitation in the manuscript if such assays are not possible.

3. In Fig. 3 and 4, the authors demonstrated that expression of CD40 (and correspondingly CD40L by CD4+ T cells) and CD70 (CD27 by CD8+ T cells) increased the proportion of stable contacts formed. Was the stopping speed (as shown in Fig. 2D) altered under such conditions compared to the pMHC only condition, e.g. did it take the antigen-specific T cells less time to arrest?

It would also be useful for the authors to comment on whether such changes occur through modifying TCR signaling threshold, or through direct mediation of actomyosin network and/or adhesion molecules (such as integrins) that enabled stronger adhesion to be formed.

4. For Fig. 5, please clarify if the localization of CD40L at the interface with CD40-expressing CHO cells was a predominant feature under the experimental condition. Quantification of the proportion of cells that exhibited such localization would be useful.

Additionally, the figure only showed up to 10 minutes of the interaction. Did the CD40L stay localized at the interface until the end of the imaging period (assuming more than an hour) or were they transient events that occurred during the early stage of contact?

Minor points:

1. Representative flow cytometry plots would be useful for Supp. Fig. 2 where the different subsets of the T cells were classified.

2. The information for the antibody used to stain for MHC-I is missing.

3. The Zenodo data repository stated in the manuscript (10.5281/zenodo.15275955) does not exist, or at least inaccessible to the reviewer. This reviewer has not been able to access the raw data or at least utilize the "Cell contacts" web tools to assess the analysis method as described.

4. In lines 250-252, the authors wrote "These findings are consistent with CD27 expression - the receptor for CD70 -which is relatively low on naïve CD8⁺ T cells and strongly upregulated after 3 and 5 days of pre-activation with antibody-coated beads (Supplementary Figure 3A, D)". This is not accurate. Although CD27 can become further upregulated upon activation, naïve CD4 and CD8 T cells have been consistently shown to express at intermediate-to high levels in mouse and human samples (refer to the works of van Lier et al., 10.1111/j.1600-065X.2009.00774.x).

5. Lines 266-268 - please rephrase "To do so, we imaged the arrest on CD40L-mCherry expressing CHO-pMHC-II cells of OT-II CD4⁺ T cells that had been pre-activated with antibody-coated beads and transduced with a retroviral backbone encoding CD40L-mCherry." as the sentence was very confusing.

A 'scan and stop' assay identifies CD40 and CD70 as selective regulators of T cell arrest on APCs

Vincent Gloe, Richard Schregle, Christoph Ratswohl and Jérémie Rossy

General Response to the Editor and Reviewers:

We would like to express our sincere gratitude to the editor and the reviewers for their time, constructive feedback, and patience during the revision of this manuscript. The comments have prompted us to perform crucial validation experiments and refine our narrative, resulting in a stronger and more rigorous study.

Most notably, following the recommendations of reviewer 1 and the editor, we have successfully incorporated neutralizing blocking antibodies into our assay. As detailed below, these experiments definitively confirm the specificity of the CD40- and CD70-mediated T cell arrest. Furthermore, we have standardized our screening panels across both CD4⁺ and CD8⁺ T cell experiments to ensure direct comparability.

Below, we provide a detailed, point-by-point response to each reviewer's comments. All changes and additions within the revised manuscript have been highlighted in yellow.

Reviewer #1 (Comments to the Authors (Required)):

This is an intriguing study revealing that costimulatory ligands CD40 and CD70 promote CD4 and CD8 T cell arrest on APCs, respectively. The study was based on coculturing OT-II or OT-I T cells with artificial antigen presenting cells: CHO cells overexpressing a single chain pMHC (for TCR activation) and either CD40 or CD70. While the conceptual message is interesting, the study can be strengthened in multiple aspect in order to be more complete and convincing. I have the following suggestions to improve the study.

Major:

1. If the T cell arrest occurs at minutes time scale, what is the rationale of measuring the effects of pre-activation and/or costimulatory ligands on different days? Wouldn't it be more informative to measure these effects over the time course of minutes or hours?

We thank the reviewer for this important question, which gives us the opportunity to clarify our experimental rationale. We recognize that the phrasing in our original manuscript may have been ambiguous.

The reviewer is correct that T cell arrest is a rapid event, occurring on a timescale of minutes, which our 'scan and stop' assay is designed to capture. The 'different days' (day 0, 3, and 5) mentioned in our experiments do not represent a time course of the arrest assay itself. Instead, this refers to the duration of T cell pre-activation that was performed *before* the cells were introduced into the assay.

Our goal was to investigate if and how the T cell's activation history influences its stopping behavior, particularly in response to co-stimulatory signals. To achieve this, we generated distinct T cell populations:

- **Day 0:** Naïve T cells, taken directly after isolation.
- **Day 3 and Day 5:** Antigen-experienced T cells, which were pre-activated with anti-CD3/CD28 beads for 3 or 5 days. This process leads to differentiation into memory-phenotype cells and alters the expression profile of key co-stimulatory receptors like CD40L and CD27.

By comparing these distinct cell populations in our short-term (90-minute) arrest assay, we could directly test how the cell's prior activation state affects its capacity to arrest upon encountering cognate pMHC with or without co-stimulatory ligands. We agree that measuring arrest after only minutes or hours of activation would be an interesting experiment, but it would address a different biological question related to the behavior of newly-activated T cells, which was outside the scope of our current study.

To prevent this confusion for future readers, we have revised the manuscript to more clearly define the term 'pre-activation' (lines 185-188) and have updated our figure axes accordingly.

2. To improve the physiological relevance of the findings, can the authors please measure the expression of the costimulatory ligands on the CHO cells using flow cytometry and compare these with those on dendritic cells?

We thank the reviewer for raising this important point. We completely agree that validating the expression levels of co-stimulatory ligands against physiological APCs is crucial for the interpretation of our findings.

This analysis was performed and included in our original submission. As shown in **Supplementary Figure 3**, we used flow cytometry to compare the surface expression of the relevant co-stimulatory ligands on our transduced CHO cells with their expression on mature, lipopolysaccharide (LPS)-stimulated bone marrow-derived dendritic cells (BMDCs). The results confirmed that the expression levels on our CHO cells are indeed comparable to those found on mature BMDCs, ensuring the physiological relevance of our system.

We apologize that the reference to this key validation was not sufficiently obvious in our original manuscript. To ensure this point is clear to all readers, we have revised the text in the results section (lines 219-22) to more explicitly describe this validation and clearly direct the reader to the data in **Supplementary Figure 3**.

3. What is the rationale of using different set of ligands for CD4 (Fig. 3) and CD8 (Fig. 4) T cells? HVEM and OX40L can be included for CD4 experiments to make it consistent with the CD8 conditions.

We thank the reviewer for raising this point regarding the consistency between our experimental panels. The reviewer is correct that in our original submission, the inclusion of HVEM and OX40L for CD8⁺ T cells but not for CD4⁺ T cells created an inconsistency.

To improve clarity, focus, and direct comparability of our findings, we have followed the reviewer's advice to make the ligand panels identical. In our revised manuscript, we have removed HVEM from the study entirely and added OX40L to the CD4⁺ T cell experiments. Now, both Figure 3 (for CD4⁺ T cells) and Figure 4 (for CD8⁺ T cells) assess the exact same panel of co-stimulatory ligands: CD80, 4-1BB-L, CD70, CD40, and OX40L.

This change streamlines the presentation and sharpens the manuscript's focus on our key discovery: the specific roles of CD40 and CD70 in regulating T cell arrest. We have updated Figures 3 and 4, as well as the corresponding text in the Results section (lines 217-218), to reflect this standardized panel.

4. Some data in Fig. 3 and Fig. 4 lack day 3 conditions. Can the authors please add the condition? Indeed, CD40 significantly increases the % stable contact on day 3. This could be true for other ligands too.

We thank the reviewer for their observation and for pointing out this omission in our original data. The reviewer is correct that the Day 3 pre-activation condition was missing for our analysis of 4-1BBL on CD4⁺ T cells, creating an incomplete picture.

Following the reviewer's suggestion, we have now performed these missing experiments. The new data for the 3-day pre-activation timepoint have been added to the revised figures. Furthermore, as noted in our response to Comment #3, we have standardized our ligand panels across both CD4⁺ and CD8⁺ experiments (removing HVEM and adding OX40L) and ensured that the Day 3 timepoints are complete for all tested ligands. The completed results confirm that ligands like 4-1BBL did not induce a significant increase in T cell arrest at the Day 3 timepoint. This is consistent with our findings for the naïve and 5-day activated conditions and strengthens our overall conclusion that the pro-arrest effects we observed are highly specific to the CD40 and CD70 pathways.

We thank the reviewer for encouraging us to complete these datasets, which has significantly improved the quality and robustness of our manuscript.

5. Fig. 5, CD40L-mCherry appears largely intracellular; is it normal or an artifact introduced by the mCherry tag? Is this possible to use a different fluorescent tag or immunostain the endogenous CD40L? Similar experiments can be done for CD70 and its receptor CD27 to make this study more complete.

We thank the reviewer for these insightful questions regarding our high-resolution imaging and the localization of the CD40L-mCherry construct.

1. Regarding the intracellular localization of CD40L: The significant intracellular pool of CD40L-mCherry we observed in migrating T cells is not an artifact but rather reflects the established cell biology of this molecule. In activated T cells, pre-existing CD40L is stored in endosomal compartments/secretory lysosomes and is rapidly mobilized to the immunological synapse upon TCR engagement (Koguchi et al, 2007). This is consistent with our data showing that 'In polarised, migrating CD4⁺ T cells, most of the CD40L signal was localised within endosomes at the rear end of the cell', which are then 'translocated to the immunological synapse' upon contact. We have added this citation in the revised manuscript to clarify this point (lines 299-300).

2. Regarding validation with immunostaining: We completely agree that validating the mCherry-tagged construct against the endogenous protein is an important control. During the revision period, we performed extensive immunostaining for endogenous CD40L on wild-type OT-II T cells co-cultured with our CHO-pMHC-II APCs. We tested two independent, well-validated anti-mouse CD40L primary antibodies (R&D Systems #AF1163 and Proteintech #16668-1-AP). Unfortunately, as shown in the **Reviewer-only Figures 1 and 2** included below, both antibodies exhibited severe, non-specific cross-reactivity with our engineered hamster (CHO) cells. The background signal on the CHO cells was comparable in intensity to the true signal on the T cells, rendering specific quantification of endogenous CD40L at the immunological synapse technically unfeasible.

Reviewer-only Figure 1A. Control staining on 3 day activated OT-II T cells

Reviewer-only Figure 1B. Staining and secondary only control on the CHO-WT & CHO pMHC-II + CD40 monolayer.

Reviewer-only Figure 2A. Control staining on 3day activated OT-II T cells and CD40L transduced CHO-WT cells.

Reviewer-only Figure 2B. Staining and secondary only control on the CHO-WT & CHO pMHC-II + CD40 monolayer.

Extended methods and observations for reviewer-only Figures 1 and 2: CD40L immunostaining in the Scan and Stop setup.

To address Reviewer 1's concern regarding the predominantly intracellular localization of CD40L–mCherry, we performed additional experiments to examine the distribution of endogenous CD40L in non-transduced, wild-type OT-II T cells under identical experimental conditions. Primary OT-II T cells were activated for three days and then co-cultured with CHO cells expressing cognate pMHC-II & CD40 for either 10 min or 90 min before fixation with 3.7% PFA and immunostaining for endogenous CD40L. We tested two independent anti-CD40L primary antibodies that have been validated in published studies, each combined with established secondary antibodies.

1. R&D Systems Antibody (#AF1163): The first antibody was a goat anti-mouse CD40L antibody previously described by Koguchi et al. (Blood, 2007). To validate this staining, we used a donkey anti-goat Alexa Fluor 488–conjugated secondary antibody. When T cells were plated alone on fibronectin-coated coverslips, we observed a faint membrane-associated signal; however, most of the staining pattern was diffuse and intracellular, indicating only partially specific binding (**Figure 1A**). For co-cultures with CHO cells, a donkey anti-goat Alexa Fluor 568–conjugated secondary antibody was used. While staining with only the secondary antibody was clean, the full staining procedure resulted in both CHO and T cells displaying comparable signal intensities, confirming substantial non-specific background binding (**Figure 1B**).

2. Proteintech Antibody (#16668-1-AP): The second antibody was a rabbit anti-mouse CD40L antibody previously cited by Yu et al. (Science Advances, 2020). For single T-cell experiments, we used an Alexa Fluor 488–conjugated secondary antibody, and for co-cultures with CHO cells, an Alexa Fluor 568–conjugated secondary antibody. This antibody combination produced a similar outcome: both CHO and T cells displayed comparable high-intensity signals, confirming substantial non-specific staining (**Figure 2A, B**).

We hypothesize that this background staining arises from cross-reactivity with an abundant hamster (CHO) cell surface protein, rendering the specific detection of endogenous CD40L at the immunological synapse unfeasible within the revision period.

Nevertheless, these data yielded an important confirmatory observation: the characteristic “pushing” phenotype we describe in the manuscript, where T cells profoundly deform and displace the cognate pMHC-II⁺ CD40⁺ CHO cells, was consistently observed in these non-transduced OT-II T cells after 90 minutes of contact (best visualized in the iRFP713 channel in **Figures 1B and 2B**). This confirms that the severe mechanical deformation of the APC is a genuine biological phenomenon and not an artifact of the CD40L-mCherry transduction.

6. The study can be benefited from testing whether the addition of blockade antibodies that disrupt either CD40-CD40L or CD70-CD27 interactions abrogate the effects observed.

We thank the reviewer and the editor for raising this critical point and for suggesting this experiment to confirm the specificity of the receptor-ligand interactions. We completely agree that the use of blocking antibodies is the gold standard for demonstrating the functional necessity of these specific pathways.

Following this suggestion, we have now performed the requested blockade experiments. We introduced neutralizing antibodies against CD40 and CD70 into our 'scan and stop' co-culture system. As now detailed in the revised Results section (lines 240-246, 274-276) and figures:

- The addition of a CD40-blocking antibody completely abrogated the enhanced stable contact formation of pre-activated CD4⁺ T cells induced by CD40-expressing APCs (**Figure 3F**).
- Similarly, disruption of the CD70-CD27 axis with a CD70-blocking antibody effectively abolished the increased arrest of CD8⁺ T cells observed on CD70-expressing APCs (**Figure 4F**).

In both cases, the blockade successfully reverted the arrest frequency back to baseline levels (comparable to the pMHC-only control conditions). These new functional data definitively confirm the specificity of our minimalistic APC system and establish that direct receptor-ligand engagement of these co-stimulatory pathways is strictly required to drive the enhanced T cell arrest.

7. Related to the point above, the antibody blockade approach could allow the authors to test using peptide loaded DCs as a more physiological APCs, and test the necessity of the CD40-CD40L or CD70-CD27 interactions in promoting the T cell stop. Their minimalistic CHO system is a nice way to test the sufficiency, but not the necessity.

We thank the reviewer for this excellent and insightful suggestion. The point they raise about our CHO system demonstrating sufficiency, while a DC-based system would be required to test necessity, is an important scientific distinction that we fully agree with.

The primary goal of our current study was precisely to overcome the inherent complexity of physiological APCs in order to ask a very specific question. Professional APCs like dendritic cells co-express a wide and variable array of co-stimulatory and co-inhibitory ligands (e.g., CD80, CD86, ICOSL, PD-L1) and secrete numerous chemokines and cytokines, all of which collectively influence T cell behavior. To test the 'necessity' of a single pathway like CD40-CD40L in such a complex system would require either extensive genetic editing of the DCs or the use of a comprehensive cocktail of blocking antibodies to neutralize all other confounding pathways. This would be a significant undertaking that would run counter to the clean, reductionist question we aimed to address.

Furthermore, the general consequences of blocking these individual pathways on DCs are already well-established. For instance, disrupting the CD40-CD40L interaction is known to profoundly inhibit DC-mediated T cell proliferation and activation (Mackey et al, 1998). Similarly, blocking the CD70-CD27 pathway impairs T cell expansion and effector function in response to DCs (Borst et al, 2005).

Our study was not designed to rediscover these established functional roles, but rather to leverage our novel assay to isolate the specific contribution of these molecules to the distinct physical process of T cell arrest, separate from other downstream outcomes.

Therefore, while we agree the proposed DC experiment is a logical future study, it represents a significant new direction beyond the defined scope of this work. In line with the editor's guidance, we have focused our revision efforts on the specifically requested validations. To acknowledge the reviewer's excellent point, we have now incorporated this context into our Discussion section, highlighting the question of 'necessity' as a key topic for future investigation (lines 399-403).

Minor:

1. *Color scheme of the bar graphs a bit confusing. For example, in Fig. 4B-G, the color of some of the bars are similar to the color of the molecules/cells in Fig. 4A. I suggest that authors use open bars to indicate pMHC-I alone condition and the filled bar to indicate +costim ligand conditions.*

We thank the reviewer for bringing this visual ambiguity to our attention. We agree that the unintentional color similarity between the schematic in Figure 4A and the data bars in panels B–G could be confusing to the reader.

To resolve this, and to maintain stylistic consistency with the bar graphs across all the other figures in our manuscript, we opted to update the color scheme of the schematic in Figure 4A so that it perfectly matches the design used in Figure 3A. This completely removes the overlapping color palettes and ensures the data presentation is now clear and unambiguous. We appreciate the reviewer's keen eye for these details.

2. *"We first transduced pMHC-expressing CHO cells with CD80, which is the ligand for CD28". I'd change "the ligand" to "a ligand" because CD80 is not the only ligand for CD28.*

The reviewer is correct, and we have modified the sentence as suggested (line 216).

3. *The rationale of testing CD70 was not clearly described.*

We thank the reviewer for pointing out that our rationale for including CD70 was not sufficiently described in the original manuscript.

Our selection of co-stimulatory molecules was guided by their established, yet distinct, roles in T cell activation. We specifically chose to include CD70 because its receptor, CD27, is known to be expressed on both naïve and activated T cells and plays a critical role in supporting the expansion and memory formation of T cells, particularly for the CD8+ subset. Given its importance in sustaining T cell responses, we hypothesized that it might also play a role in an earlier step: the initial physical arrest of the T cell on the APC. Our results subsequently confirmed this hypothesis.

To make our reasoning clear to the reader, we have now added a sentence and citations to the Results section of the revised manuscript that explicitly states this rationale for including CD70 in our screening panel (lines 261-267).

Reviewer #2 (Comments to the Authors (Required)):

The manuscript titled "'scan and stop' assay identifies CD40 and CD70 as selective regulators of T cell arrest on APCs" by Gloe et al. utilizes a well-established property of arresting when encountering an agonist antigen to

identify co-stimulatory signals required to achieve an arrest. Indeed, through a number of intravital imaging studies, the arrest of T cells (or "stop signal") has been established as a hallmark of T cell activation. Gloe et al. leverage this property nicely to generate a controlled ex vivo assay for identifying co-stimulatory regulators of arrest. For this, develop a cellular antigen presentation system that has minimal background engagements, thereby revealing quantitative information on phenotype resulting from the co-stimulatory signal alone. The assay system is neat and provides a crucial testing ground for the role of surface receptors in a more physiologically relevant mechanical context than possible with most planar ligand reconstitution systems that are conventionally used for such assays. Overall, their system is clean, experiments are well constructed and presented with appropriate controls, and the manuscript is also written very well. I highly recommend it for the publication.

My only suggestions are:

1. The authors identify CD70 as a surprising ligand inducing arrest. They have nicely discussed the known literature on CD70. However, more discussion on what it may be doing is missing. This is best discussed, or at least speculated, in the context of synapse formation and breaking kinetics. For instance, does CD70 promote entry into arrest, or prolong the arrest phase (e.g., Kumari et al. EMBO J., 2020), or both? Perhaps authors could add CD70 neutralizing antibody after initial arrest and examine if the arrest is broken, and T cells start migrating again to address this?

We are very grateful to the reviewer for their positive feedback and for this excellent suggestion, which prompted us to think more deeply about the mechanistic interpretation of our findings.

The reviewer correctly identifies two potential mechanisms: promoting entry into arrest versus prolonging the arrest phase. This is a critical distinction. However, given the design of our assay, where a 'stable contact' is defined as one that persists until the end of our 90-minute acquisition, our readout is a measure of the probability that a T cell will commit to a stable, long-term arrest. Our current assay cannot, therefore, distinguish between a mechanism that promotes the initial *decision* to arrest versus one that *prolongs* an arrest that has already been initiated, as both would lead to a higher percentage of cells being scored as 'stable'.

That being said, we believe the data point toward the first mechanism, promoting the initial commitment to arrest. We did not observe a significant change in the number of *transient* contacts in the presence of CD70 (data not shown), suggesting that CD70 signaling is not merely making short interactions last slightly longer, but is rather increasing the likelihood that a T cell will form a stable synapse in the first place.

We have clarified this in our revised Discussion (lines 394-399). We now propose that CD70-CD27 signaling likely lowers the activation threshold required for a T cell to commit to a stable synapse, thereby increasing the probability of arrest upon antigen encounter. We thank the reviewer, as their comment has helped us to refine and clarify the interpretation of our results.

2. Following up on the previous point, the authors' assay system is uniquely suited to address the many mechanical dimensions of receptor function at the synaptic interface as well as overall synapse evolution and lifetime. In addition, the manuscript dedicates great effort to setting up and characterizing the system itself. It will be great if authors can devote a paragraph in the discussion to highlight the broad questions in T cell biology that their methodology can address in the future.

We sincerely thank the reviewer for their enthusiasm for our assay system and for this great suggestion. We agree that highlighting the future potential of our method would be a valuable addition to the manuscript's conclusion.

As suggested, we have added a new paragraph at the end of our Discussion section (lines 409-420). In this section, we outline several broad questions in T cell biology that our 'scan and stop' assay is uniquely suited to address. This includes investigating the mechanical dimensions of synapse formation by using optogenetic tools to dynamically control APC stiffness and exploring how the broader microenvironment influences T cell scanning by engineering the surrounding non-activating CHO cells to recreate aspects of a lymph node or tumor microenvironment. We believe this new paragraph strengthens the manuscript's outlook, and we are grateful for the reviewer's encouragement to include it

Reviewer #3 (Comments to the Authors (Required)):

In this manuscript, Gloe et al. designed an in vitro T-APC assay (stop and scan) that leverages CHO cells engineered to express pMHC complex as well as selected co-stimulatory molecules to assess the role of these molecules in the arrest of antigen-specific T cells with their antigen-presenting cells. While not particularly novel, this method provides a useful platform for reductionistic evaluation of molecules that control T-APC interactions. Using this approach, the authors identified CD40L on CD4+ T cells and CD27 on CD8+ T cells - both known co-stimulatory molecules - as having potentially novel roles in mediating T cell arrest during priming with APCs.

The manuscript is well written, and the experiments were, for the most part, well controlled. The discussion section also addressed the limitations and some of the questions that this reviewer had while reading through the manuscript.

Specific points that need to be addressed:

1. In Fig. 1D-E and Fig. 2, the authors showed that the formation of "stable contact" as well as the arrest parameters were not different between naïve and Day 3 and Day 5 pre-activated T cells. Were the TCR/CD3 expression level of the pre-activated T cells different (e.g. downregulated) compared to naïve T cells? Downregulation of TCRs during activation can influence binding avidity with pMHC and consequently the T cell arrest behavior with APC, and the TCR levels should at least be measured to address whether the avidity affects T-APC interaction.

We thank the reviewer for raising this important point and for their suggestion to investigate the TCR expression levels on our pre-activated T cells. The reviewer correctly notes that TCR expression can be modulated by activation, which could in turn influence T cell arrest behavior. Following the reviewer's valuable suggestion, we performed flow cytometry to measure the surface TCR V α 2 levels on naïve T cells versus those pre-activated for 3 and 5 days. Surprisingly, we found that contrary to the canonical view of TCR downregulation that occurs in the initial hours of activation, T cells at 3 and 5 days of stimulation exhibited a higher surface level of TCR compared to naïve/resting T cells.

This new data, which we have now included in **Supplementary Figure 2D and E**, leads us to conclude that the absolute surface level of TCR is not the decisive factor for the initial decision to arrest in our system. This conclusion is supported by two observations from our data:

1. Despite having higher TCR levels, the 3-day and 5-day pre-activated T cells show the same propensity to form stable contacts as naïve T cells.
2. Our new data also show that OT-I (CD8+) T cells consistently express higher levels of TCR than OT-II (CD4+) T cells, yet both populations displayed a similar probability of arresting on their respective pMHC-bearing CHO cells.

This finding is consistent with literature showing the remarkable sensitivity of the TCR, which can be triggered by as few as a single peptide-MHC complex (Irvine et al, 2002; Huang et al, 2013). Given this high sensitivity, it is plausible that once a threshold of TCR expression is met, other factors, such as the co-stimulatory signals we investigate, become the key regulators of the final decision to commit to a stable synapse.

We have added this new data to the supplementary figures and have incorporated this into the revised manuscript (lines 195-200). We are very grateful to the reviewer for prompting this experiment, which has added a new and important layer of insight to our study.

2. Related to the above point, it is unclear if the transduced levels of single-chain OVA-pMHC in CHO cells are within physiological range or if excess amounts of pMHC signals were being presented to the antigen-specific T cells coming into contact.

Although the authors showed that the detected level of MHC-I was similar between BMDCs and CHO cells (Supp. Fig. 1G), note that unless the OVA-pMHC level was measured (e.g. using 25D1 antibody for SIINFEKL and similar OVA 323-339 antibody for pMHC-II), it is difficult to assess if the antigen level being presented is within physiological range. We may assume that all of the mouse MHC molecules detected on CHO cells were the

transduced single-chain OVA peptide-MHC complexes, and if these cells express similar levels of MHC-I and MHC-II as BMDCs, it would amount to that all of the MHC molecules expressed by BMDCs were presenting the OVA antigens.

As such, excessive amount of antigen could overcome TCR downregulation (see point #1 above) and the drop in TCR avidity, and therefore obfuscates the T cell arrest dynamics between naïve and activated T cells that could have been different under physiological contexts.

The authors can control for this by staining for the OVA peptide-MHC complexes on the engineered CHO cells and compare them to those measured on BMDCs incubated with a reasonable amount of OVA antigen where the presented antigens have gone through antigen processing steps under physiological conditions, or at least address this limitation in the manuscript if such assays are not possible.

We thank the reviewer for this insightful comment, which gives us an excellent opportunity to further clarify the specific nature of our experimental system and rationale.

The reviewer correctly points out the importance of measuring the specific peptide-MHC complex. In our system, we addressed this by engineering our CHO cells to express a single-chain pMHC construct, where the specific OVA peptide is covalently linked to the MHC molecule. Therefore, unlike in physiological APCs, essentially all of the surface MHC molecules detected on our cells are the specific, cognate pMHC complex. For this reason, measuring total surface MHC is a direct and appropriate measure of the specific pMHC being presented in our system.

We agree that this uniform, high density of a single pMHC is likely supraphysiological compared to endogenous antigen presentation. However, this is a common feature of many widely-used *in vitro* immunological assays that rely on peptide-pulsed APCs. In that standard method, high concentrations of an exogenous peptide displace a large fraction of the diverse, endogenous peptides from surface MHC molecules, likely resulting in a similarly high surface density of the specific pMHC

Crucially, this high-density presentation was an intentional part of our design. We engineered the system to provide a robust and saturating TCR 'Signal 1' to specifically and cleanly interrogate the contribution of co-stimulatory molecules ('Signal 2') to the physical process of T cell arrest. As the reviewer astutely suggests, this strong Signal 1 likely explains why we did not observe significant differences in arrest between naïve and pre-activated T cells, as it can overcome more subtle variations in TCR avidity.

We have revised our Discussion to incorporate this detailed clarification (lines 333-337) and thank the reviewer for prompting us to elaborate on these important methodological aspects of our study.

3. In Fig. 3 and 4, the authors demonstrated that expression of CD40 (and correspondingly CD40L by CD4+ T cells) and CD70 (CD27 by CD8+ T cells) increased the proportion of stable contacts formed. Was the stopping speed (as shown in Fig. 2D) altered under such conditions compared to the pMHC only condition, e.g. did it take the antigen-specific T cells less time to arrest?

It would also be useful for the authors to comment on whether such changes occur through modifying TCR signaling threshold, or through direct mediation of actomyosin network and/or adhesion molecules (such as integrins) that enabled stronger adhesion to be formed.

We thank the reviewer for these excellent questions regarding the dynamics and mechanism of co-stimulation-mediated T cell arrest.

1. Regarding the stopping speed: Following the reviewer's suggestion, we have analyzed the stopping speed for all co-stimulatory conditions. Our analysis revealed that while CD40 and CD70 increased the *probability* of stable contact, none of the ligands altered the *stopping speed* of the cells that did arrest. This finding suggests that the primary role of co-stimulation here is to lower the threshold for *committing* to a stable synapse, rather than changing the kinetics of the stop itself. We have added this complete analysis to the revised Figures 3 and 4 and modified the main text accordingly (lines 235-236, 272-273)

2. Regarding the mechanism: The reviewer asks whether the changes occur through modifying the TCR threshold or through direct mediation of adhesion molecules. Our system is ideally suited to dissect this. Our CHO cells do not express mouse adhesion molecules. While they do express endogenous hamster ICAM-1, its low sequence identity (72%) to mouse ICAM-1 makes a significant cross-reactive binding to murine LFA-1 unlikely. Therefore, in the absence of a strong integrin-adhesion axis, the arrest we observe is driven primarily by the pMHC-TCR interaction and the specific co-stimulatory signals we introduce. This supports a model where co-stimulation through CD40/CD70 lowers the TCR signaling threshold required to initiate an arrest program, which then drives the necessary cytoskeletal changes. We have clarified this in our revised Discussion (lines 346-352).

4. For Fig. 5, please clarify if the localization of CD40L at the interface with CD40-expressing CHO cells was a predominant feature under the experimental condition. Quantification of the proportion of cells that exhibited such localization would be useful.

Additionally, the figure only showed up to 10 minutes of the interaction. Did the CD40L stay localized at the interface until the end of the imaging period (assuming more than an hour) or were they transient events that occurred during the early stage of contact?

We thank the reviewer for these specific questions regarding the spatiotemporal dynamics of CD40L.

1. Regarding quantification at the interface: As detailed in our response to Reviewer 1, Point 5 (and shown in the accompanying Reviewer-Only figures), our attempts to quantify the proportion of cells exhibiting endogenous CD40L localization were unfortunately hindered by severe, non-specific cross-reactivity of the anti-mouse CD40L antibodies with our hamster (CHO) cell line. However, our new blocking antibody data (**Figure 3F**) functionally confirm that specific CD40-CD40L engagement is the predominant driver of the enhanced arrest observed in our assay.

2. Regarding long-term imaging of CD40L: The reviewer raises an excellent point about the duration of CD40L localization. In our experimental setup, continuous, high-resolution confocal z-stack imaging of primary T cells over a 90-minute period resulted in severe phototoxicity and fluorophore bleaching. This precluded the direct, continuous visualization of CD40L dynamics over the full duration of the arrest. However, because our low-magnification SaS assay demonstrates that the T cells maintain a stable, firmly arrested state for the entire 90-minute observation window—and because this arrest is completely abolished by CD40 blockade—our functional data strongly imply that receptor engagement is maintained continuously throughout the arrest phase, rather than being a strictly transient early event."

Minor points:

1. Representative flow cytometry plots would be useful for Supp. Fig. 2 where the different subsets of the T cells were classified.

We thank the reviewer for this helpful suggestion to improve our supplementary data. As requested, we have now added a new panel A to **Supplementary Figure 2** that shows representative flow cytometry plots. These plots illustrate our gating strategy for identifying naïve (CD44-CD62L+), central memory (CD44+CD62L+), and effector/effector memory (CD44+CD62L-) T cell subsets at the different time points of pre-activation. This provides a clearer view of our primary data.

2. The information for the antibody used to stain for MHC-I is missing.

We thank the reviewer for spotting this omission. The reviewer is correct that we had inadvertently left out the details for the anti-MHC-I antibody from our methods section. We

have now added this information to the 'Flow cytometry' subsection of the revised manuscript (lines 597-598).

3. *The Zenodo data repository stated in the manuscript (10.5281/zenodo.15275955) does not exist, or at least inaccessible to the reviewer. This reviewer has not been able to access the raw data or at least utilize the "Cell contacts" web tools to assess the analysis method as described.*

We thank the reviewer for raising this point. Our standard procedure is to make the full dataset publicly available on the Zenodo repository upon formal acceptance of the manuscript, at which point the permanent DOI listed in the manuscript becomes active.

However, we understand and appreciate the reviewer's desire to access the raw data and analysis tools during the review process to fully assess our methodology. To facilitate this, we have now made the complete dataset available through this link for the editors and reviewers: https://zenodo.org/uploads/15275955?token=eyJhbGciOiJIUzUxMiJ9.eyJpZCI6IjJiOGEyMDk0LTIyYmYyMmNGQxMi1iNDEzLTM0ZTZkYjU2ZDEzNiIsImRhdGEiOnt9LCJyYW5kb20iOiJmODFiNDViZDQ4MzUyN2ZiOGI4YzU1MDNmOGM4MmY3NCJ9.FFfz5IfV1wns6wPAloG8H-89Q5mHZxyOHRU8hJ4X0xsZfKVeQ3ajyQyfEAFb2b3_dYye42cW-J9dd9Tb8Mer3A

We thank the reviewer for their diligence and for prompting us to make the data available at this earlier stage."

4. *In lines 250-252, the authors wrote "These findings are consistent with CD27 expression - the receptor for CD70 -which is relatively low on naïve CD8⁺ T cells and strongly upregulated after 3 and 5 days of pre-activation with antibody-coated beads (Supplementary Figure 3A, D)". This is not accurate. Although CD27 can become further upregulated upon activation, naïve CD4 and CD8 T cells have been consistently shown to express at intermediate-to high levels in mouse and human samples (refer to the works of van Lier et al., 10.1111/j.1600-065X.2009.00774.x).*

We thank the reviewer for pointing out this inaccuracy. The reviewer is absolutely correct; our description of CD27 expression as 'relatively low' on naïve T cells was imprecise and misleading. While our data show a strong upregulation upon activation, we acknowledge that naïve T cells do express functional levels of CD27.

We have now corrected this sentence in the revised manuscript (lines 269-271) to more accurately reflect the established literature and our own data.

5. *Lines 266-268 - please rephrase "To do so, we imaged the arrest on CD40L-mCherry expressing CHO-pMHC-II cells of OT-II CD4⁺ T cells that had been pre-activated with antibody-coated beads and transduced with a retroviral backbone encoding CD40L-mCherry." as the sentence was very confusing.*

We thank the reviewer for pointing out this confusing and poorly phrased sentence. The reviewer is correct that the sentence was unclear and contained an error regarding which cell expressed CD40L-mCherry. We have now completely rewritten this sentence for clarity and accuracy in the revised manuscript (lines 294-296).

References

Borst J, Hendriks J, Xiao Y (2005) CD27 and CD70 in T cell and B cell activation. *Current Opinion in Immunology* 17: 275–281. doi:10.1016/J.COI.2005.04.004.

Huang J, Brameshuber M, Zeng X, Xie J, Li Q, Chien Y, Valitutti S, Davis MM (2013) A single peptide-major histocompatibility complex ligand triggers digital cytokine secretion in CD4(+) T cells. *Immunity* 39: 846–857. doi:10.1016/j.immuni.2013.08.036.

Irvine DJ, Purbhoo MA, Krogsgaard M, Davis MM (2002) Direct observation of ligand recognition by T cells. *Nature* 419: 845–849. doi:10.1038/nature01076.

Koguchi Y, Thauland TJ, Slifka MK, Parker DC (2007) Preformed CD40 ligand exists in secretory lysosomes in effector and memory CD4+ T cells and is quickly expressed on the cell surface in an antigen-specific manner. *Blood* 110: 2520–2527. doi:10.1182/blood-2007-03-081299.

Mackey MF, Barth RJ, Noelle RJ (1998) The role of CD40/CD154 interactions in the priming, differentiation, and effector function of helper and cytotoxic T cells. *J Leukoc Biol* 63: 418–428. doi:10.1002/jlb.63.4.418.

May 19, 2026

RE: Life Science Alliance Manuscript #LSA-2025-03401R

Dr. Jérémie Rossy
Institute of Cell Biology and Immunology Thurgau
Unterseestraße 47
Kreuzlingen s 8280
Switzerland

Dear Dr. Rossy,

Thank you for submitting your revised manuscript entitled "A 'scan and stop' assay identifies CD40 and CD70 as selective regulators of T cell arrest on APCs". We returned this work to Reviewers 1 and 3 who are now satisfied with no major concerns. We invite you to address the suggestion of Reviewer 3 to contextualize these results in view of pMHC overexpression, in the manner of your choice. We would be happy to publish your paper in Life Science Alliance pending this change and final revisions necessary to meet our formatting guidelines.

MANUSCRIPT ORGANIZATION AND FORMATTING:

To avoid unnecessary delays in the acceptance and publication of your paper, please read the following information carefully. Full guidelines are available on our Instructions for Authors page, <https://www.life-science-alliance.org/authors>

- Please add your main, supplementary figure, table, and video legends to the main manuscript text after the references section.
- Please add the X and Bluesky handles of your host institute/organization, as well as your own, and/or one of the authors, in our system.
- Please mark the Corresponding Author on the title page of the manuscript file.
- Please add affiliations for the authors on the title page of the manuscript. Numbers in superscript should be used to indicate the department, institution, city, and country for each author.
- Since Figure S3 has only one panel, it is unnecessary to label it as A. Please remove it from the figure and its legend.
- Please add a Conflict of Interest statement to your main manuscript text.
- Please add a callout for Figure S2C to your main manuscript text.
- Please include details on microscopy illumination and temperature in the Methods section.
- Please expand the methods details on lines 547-8 to describe the HTML GUI.
- We appreciate the separate file describing scMHCII cloning, and while you may leave this as a supplementary file, we suggest this may be incorporated into the main methods section and the associated figures may be added as supplementary figures. However we leave this change to your discretion.

We welcome submissions of potential cover images for the issue of LSA in which your work would appear. If you have high quality images associated with this work, please feel free to email these, with a caption, to the journal office.

LSA encourages authors to provide a 30-60 second video where the study is briefly explained. These videos will be appear embedded with the manuscript online at Life Science Alliance, and on social media to promote the published paper and authors (for examples, see <https://docs.google.com/document/d/1-UWCfbE4pGcDdcgzcmiuJl2XMBJnxKYeqRvLLrLSo8s/edit?usp=sharing>). Corresponding or first-authors are welcome to submit the video. Please submit only one video per manuscript. The video can be emailed to contact@life-science-alliance.org

FINAL FILES:

The following items are required for acceptance.

The license to publish form must be signed before your manuscript can be sent to production. A link to the license to publish form will be available to the corresponding author only. Please take a moment to check your funder requirements.

Thank you for your attention to these final processing requirements. Please revise and format the manuscript and upload materials as soon as you are able.

Thank you for this interesting contribution to the literature. We look forward to publishing your paper in Life Science Alliance.

Sincerely,

Reviewer #1 (Comments to the Authors (Required)):

I thank the authors for their efforts in addressing my prior concerns. The manuscript has been substantially strengthened, and I support its publication.

Reviewer #3 (Comments to the Authors (Required)):

The authors have satisfactorily answered all of my questions.

However, given that the transduced pMHC-CHO system used in this paper likely presented supraphysiological and saturating pMHC (Signal 1) to the T cells, while this can be seen as an advantage to disentangle the influences of Signal 1 from Signal 2, the authors should at least address that this is also a limitation of the assay, as high level of Signal 1 under in vivo setting can affect the dynamics of T cell arrest with their antigen-presenting cells, and thus also the downstream kinetics of molecular interactions involved. As such, while this paper has presented a minimalistic system to investigate the role of co-stimulatory molecules in T cell arrest, this limitation should be pointed out clearly in the text that extrapolation of findings obtained using such method should be carefully considered and requires further scrutiny under more physiological experimental setting, both in vitro and in vivo.

Other than that, this is a well-written paper that is suitable for publication.

May 26, 2026

RE: Life Science Alliance Manuscript #LSA-2025-03401RR

Dr. Jérémie Rossy
Institute of Cell Biology and Immunology Thurgau
Unterseestraße 47
Kreuzlingen s 8280
Switzerland

Dear Dr. Rossy,

Thank you for submitting your Research Article entitled "A 'scan and stop' assay identifies CD40 and CD70 as selective regulators of T cell arrest on APCs". It is a pleasure to let you know that your manuscript is now accepted for publication in Life Science Alliance. Congratulations on this interesting work.

Your article will publish open access upon publication under a CC-BY license.

DISTRIBUTION OF MATERIALS:

Again, congratulations on a very nice paper. I hope you found the review process to be constructive and are pleased with how the manuscript was handled editorially. We look forward to future exciting submissions from your lab.

Sincerely,
